# Dual roles of EGO-1 and RRF-1 in regulating germline exo-RNAi efficiency in *Caenorhabditis elegans*

Katsufumi Dejima (ID), Keita Yoshida (ID) & Shohei Mitani (ID) ✉

## Abstract

**RNA interference (RNAi) is widely used in life science research and is critical for diverse biological processes, such as germline development and antiviral defense. In the germline of *Caenorhabditis elegans*, exogenous RNAi (exo-RNAi), the RNA-dependent RNA polymerases EGO-1 and RRF-1 play redundant roles in facilitating small RNA amplification. However, their coordination during the regulation of exo-RNAi processes in the germline remains unclear. Here, we examine non-null mutants of the *ego-1* gene and find that *ego-1(S1198L)* animals exhibit germline exo-RNAi defects with normal fertility, abnormalities in germ granules, and synthetic temperature-dependent sterility with *rrf-1*. The exo-RNAi defects in *ego-1(S1198L)* are partially restored by inhibiting *hrde-1* and *znfx-1*. Germline exo-RNAi defects are observed in wild-type and *ego-1(S1198L)* heterozygous descendants derived from *ego-1(S1198L)*, but these are suppressed by ancestral inhibition of *rrf-1*. Our data reveal a dual role for EGO-1 in the positive regulation of germline exo-RNAi: it not only mediates target silencing through its RNA-dependent RNA polymerase activity, but also licenses exo-RNAi gene expression, which is antagonized by RRF-1.**

**Keywords** *Caenorhabditis elegans*; RNA-Dependent RNA Polymerase; RNA Interference; Small RNA; Epigenetics
**Subject Category** RNA Biology

## Introduction

RNA interference (RNAi) is a process in which double-stranded RNA (dsRNA) triggers the silencing of mRNAs with sequence homology to the dsRNA (Fire et al, 1998). RNAi plays a role in the regulation of gene expression, RNA quality control, and anti-transposon and antiviral responses in many eukaryotes, as well as in epigenetic inheritance in certain eukaryotes (Castel and Martienssen, 2013). RNAi can be induced using endogenous or exogenous dsRNA sources. Regardless of whether initiated by an exogenous or endogenous source, RNAi in *Caenorhabditis elegans* is a common process that leads to the amplification of secondary small RNAs (sRNAs) by RNA-dependent RNA polymerases

(RdRPs) (Pak and Fire, 2007). In exogenous RNAi (exo-RNAi), the endonuclease Dicer cleaves the exogenously introduced long dsRNAs into primary sRNAs (Tabara et al, 2002). The primary Argonaute protein RDE-1 loads the resulting sRNAs and recruits RdRPs to produce secondary sRNA populations that regulate their target mRNAs (Sijen et al, 2001; Tabara et al, 1999). In endogenous RNAi, several primary Argonaute proteins, including CSR-1, PRG-1, ERGO-1, and ALG-3/4, selectively bind endogenous non-coding RNAs such as 21U-RNA and 26 G RNA (Batista et al, 2008; Claycomb et al, 2009; Conine et al, 2010; Das et al, 2008; Gent et al, 2010; Pavelec et al, 2009; Ruby et al, 2006; Wang and Reinke, 2008). The 21U-RNA is transcribed from the 21U-RNA locus by RNA polymerase II under the regulation of FKH transcription factors, whereas 26G RNA is a type of secondary RNA synthesized by the enhanced RNAi (ERI) complex using spliced mRNA as a template (Cecere et al, 2012; Kennedy et al, 2004). The binding process between primary Argonaute proteins and these sRNAs facilitates targeting of the corresponding RNAs. This interaction recruits the RdRP protein, leading to the amplification of secondary sRNAs. There are four RdRPs in *C. elegans*, including EGO-1, RRF-1, RRF-2, and RRF-3 (Sijen et al, 2001; Smardon et al, 2000), all of which are involved in sRNA synthesis in the endogenous RNAi pathway. RRF-2 and RRF-3 are involved in the amplification of risi-RNA and 26G-sRNA, respectively (Simmer et al, 2002; Zhou et al, 2017). EGO-1 mediates both CSR-1 and WAGO 22 G RNA amplification, whereas RRF-1 mediates WAGO 22G RNA and risi-RNA amplification (Claycomb et al, 2009; Gu et al, 2009; Zhou et al, 2017).

In germline exo-RNAi, an intrinsic amplification system mediated by EGO-1 and RRF-1 enables efficient gene knockdown. RRF-1 functions in both germline and somatic cells, whereas EGO-1 functions exclusively in germline cells, with partial involvement in some somatic cells (Kumsta and Hansen, 2012). The *ego-1* and *rrf-1* genes are physically close in the genome and are likely transcribed together as an operon. Thus, they may exhibit similar epigenetic and transcriptional regulations. However, it is likely that their functional differences are not solely due to transcriptional regulation but rather arise from the specific roles played by each molecule. While RRF-1 has the ability to synthesize mutator complex-dependent sRNA, which is critical for gene silencing in exo-RNAi, EGO-1 plays a compensatory role in promoting germline exo-RNAi, complementing the function of RRF-1 (Claycomb et al, 2009; Gu et al, 2009; Zhou et al, 2017). This idea is supported by two key findings: (i) *rrf-1* null mutants are exo-RNAi sensitive in the germline (Sijen et al, 2001); and (ii) in the

Department of Physiology, Tokyo Women's Medical University School of Medicine, 8-1, Kawada-cho, Shinjuku-ku, Tokyo 162-8666, Japan. ✉E-mail: mitani.shohei@twmu.ac.jp

absence of RRF-1, EGO-1 is recruited to the mutator complex to compensate for the loss of RRF-1 function (Phillips et al, 2012; Phillips and Updike, 2022). However, ego-1 null mutants exhibit defects in exo-RNAi targeting germline genes, whereas rrf-1 mutants do not, and the exact reason for this difference remains unclear. Null mutations in ego-1 result in severe germline developmental abnormalities (Smardon et al, 2000; Vought et al, 2005), making it challenging to conduct detailed analyses, including assessment of exo-RNAi effects.

Understanding the role of germ granules is crucial for studying germline exo-RNAi. P granules safeguard the transcripts of germline-expressing genes against silencing mediated by PRG-1 and HRDE-1/WAGO-9 (Dodson and Kennedy, 2019; Ouyang et al, 2019). This protective mechanism is important for exo-RNAi activity, as the safeguarded transcripts include the RNAi genes sid-1 and rde-11 (Dodson and Kennedy, 2019; Lev et al, 2019; Ouyang et al, 2019). Transcripts of sid-1 and rde-11 accumulate in P granules, and the loss of P granules results in their dispersion and aberrant levels of WAGO 22 G RNA associated with HRDE-1 (Dodson and Kennedy, 2019; Ouyang et al, 2019). In turn, this silence causes defects in germline exo-RNAi. In contrast, CSR-1 opposes PRG-1/piRNA complex activity, which triggers HRDE-1-mediated transgenerational gene silencing (Shen et al, 2018). Therefore, as both WAGO and CSR-1 class sRNAs are involved, EGO-1 and RRF-1 seem to have differential roles in synthesizing the sRNAs loaded onto these Argonauts. Disruption of the E granule, of which EGO-1 is a component, has recently been shown to upregulate sRNA targeting sid-1 and rde-11, leading to silencing of these genes (Chen et al, 2024). However, the orchestration of these complex regulatory mechanisms in germline exo-RNAi remains unknown.

Here, we examined non-null alleles of the ego-1 gene from the Million Mutation Project (MMP) collection (Thompson et al, 2013). Four alleles showed germline exo-RNAi defects (Rde) with normal fertility at 20 °C, in contrast to ego-1 null animals. We further analyzed the mutant strain with ego-1(S1198L) and found synthetic effects on the temperature-sensitive sterile phenotype of ego-1(S1198L) by rrf-1 deletion mutation. The levels of HRDE-1::GFP increased in ego-1(S1198L) pachytene-stage cells. In addition, we found that sid-1 and rde-11 transcripts in ego-1(S1198L) were downregulated, and this effect was suppressed in hrde-1 mutants. The Rde phenotype persists over generations in an RRF-1-dependent manner, even in the absence of the original ego-1(S1198L) mutation. Our data demonstrate that EGO-1 plays a role in enhancing the robustness of exo-RNAi in the germline by mediating at least two processes: it acts as an RdRP that is necessary for target gene silencing by complementing RRF-1, and it regulates the expression of exo-RNAi genes in the germline. These findings reveal that an extensive interdependent RdRP network is responsible for regulating germline exo-RNAi.

# Results

## Some non-null ego-1 alleles show germline exo-RNAi defects

Null mutants of the RdRP gene, ego-1, exhibit a completely sterile phenotype (Smardon et al, 2000). Within the MMP strain collection, 23 viable and fertile strains have been identified, each carrying a missense mutation at the ego-1 locus (Table 1; Fig. EV1) (Dejima and Mitani, 2022; Thompson et al, 2013). Single amino acid substitutions were distributed throughout the EGO-1 protein, including the RdRP and coiled-coil domains (Fig. 1A). Bioinformatic programs commonly used to predict the effects of missense mutations suggested that some mutations were likely to have deleterious effects (Table 1) (Adzhubei et al, 2010; Choi et al, 2012; Ng and Henikoff, 2001). Given the critical role of the EGO-1 protein in RNAi, we investigated whether germline exo-RNAi was defective in all mutants. To investigate this phenotype, we fed these ego-1 mutant hermaphrodites with bacteria expressing dsRNA against pos-1 or pop-1. The pos-1 and pop-1 genes are maternally expressed in germ cells and are essential for embryonic viability. Among the mutant strains that were outcrossed with the fluorescent chromosomal balancer tmC18[tmIs1200] (Dejima and Mitani, 2022), ego-1(gk357146), ego-1(gk426642), ego-1(gk532049), and ego-1(gk882383) showed an Rde phenotype when they were fed HT115 bacteria expressing dsRNA specific to germline genes, either pos-1 or pop-1 dsRNA (Table 1; Fig. 1B,C). Hereafter, we refer to ego-1(gk357146) as ego-1(V1128E), ego-1(gk426642) as ego-1(R539Q), ego-1(gk532049) as ego-1(S1198L), and ego-1(gk882383) as ego-1(C823Y).

## Mutants with the exo-RNAi defective ego-1 allele are fertile but show a temperature-sensitive sterile phenotype in the absence of RRF-1

All of these germline exo-RNAi defective mutants showed normal brood size at 20 °C, whereas the ego-1(tm521) deletion mutant showed a completely sterile phenotype (Fig. 1D) (Ketting et al, 1999). Next, we examined the brood size of these mutants at high temperatures (Fig. 1E). The ego-1 mutants that exhibited the strong germline exo-RNAi defective phenotype (R539Q and S1198L) showed mild defects in brood size at 25 °C. In contrast, ego-1(V1128E) and ego-1(C823Y), which have weak germline exo-RNAi defects, showed a relatively large reduction in brood size at 25 °C. The lack of correlation suggests that germline exo-RNAi activity in these mutants is independent of gamete production at high temperatures. For further analysis, we focused on ego-1(S1198L), which exhibited the most pronounced Rde phenotype. To investigate the functional relevance of rrf-1, we knocked out the rrf-1 gene, which is adjacent to the ego-1 gene in the genome, in the ego-1(S1198L) background using the CRISPR/Cas9 system to create rrf-1(tm9941) ego-1(S1198L) double mutants (Fig. 2A). For comparison, we generated a similar deletion allele, rrf-1(tm9951) (Fig. 2A). While ego-1(S1198L) was sensitive to somatic RNAi, rrf-1(tm9951) and rrf-1(tm9941) ego-1(S1198L) mutants were completely defective in somatic RNAi (Fig. EV2A,B). The rrf-1 single deletion mutant rrf-1(tm9951), as well as a commonly used and larger deletion allele rrf-1(pk1417) (Sijen et al, 2001), showed normal brood size at both 20 and 25 °C (Fig. 2B,C). In contrast, rrf-1(tm9941) ego-1(S1198L) double-mutant homozygotes showed temperature-sensitive sterility similar to mut-14(pk738) and mut-15(tm1358) mutants (Fig. 2B,C), suggesting that thermotolerant gametogenesis in ego-1(S1198L) is maintained by a redundant function of RRF-1 in an endogenous RNAi mechanism mediated by the mutator complex.

**Table 1.** Effects of *ego-1* MMP mutations on germline exogenous RNAi response.

| Allele name | Amino acid change | Amino acid aligned in RRF-1 | PolyPhen-2 | SHIFT | PROVEAN | *pos-1* RNAi (if not Rde, % Emb ± SD is shown) | *pop-1* RNAi (if not Rde, % Emb ± SD is shown) |
|---|---|---|---|---|---|---|---|
| *gk115401* | E20K | S | benign | NA | neutral | 100 ± 0 | 100 ± 0 |
| *gk720210* | V53I | I | benign | NA | neutral | 100 ± 0 | 100 ± 0 |
| *gk470185* | P161S | - | benign | NA | neutral | 100 ± 0 | 100 ± 0 |
| *gk498425* | L246F | V | probably damaging | NA | Deleterious | 98.3 ± 1.5 | 100 ± 0 |
| *gk115397* | V277M | C | possibly damaging | NA | neutral | 100 ± 0 | 100 ± 0 |
| *gk115395* | D304N | H | benign | NA | neutral | 100 ± 0 | 100 ± 0 |
| *gk837385* | Y313F | Y (conserved) | probably damaging | NA | neutral | 100 ± 0 | 100 ± 0 |
| *gk115394* | R341K | S | benign | NA | neutral | 100 ± 0 | 100 ± 0 |
| *gk426642* | R539Q | R (conserved) | possibly damaging | TOLERATED | Deleterious | Rde (data is shown in Fig. 1B,C) | Rde (data is shown in Fig. 1B,C) |
| *gk896494* | L721F | I | benign | TOLERATED | neutral | 96.7 ± 5.8 | 100 ± 0 |
| *gk115393* | R722H | R (conserved) | benign | DELETERIOUS | neutral | 100 ± 0 | 100 ± 0 |
| *gk481348* | S812F | S (conserved) | benign | DELETERIOUS | Deleterious | 100 ± 0 | 100 ± 0 |
| *gk882383* | C823Y | C (conserved) | probably damaging | TOLERATED | Deleterious | Rde (data is shown in Fig. 1B,C) | Rde (data is shown in Fig. 1B,C) |
| *gk721963* | G844E | G (conserved) | probably damaging | DELETERIOUS | Deleterious | 100 ± 0 | 100 ± 0 |
| *gk357146* | V1128E | V (conserved) | possibly damaging | TOLERATED | Deleterious | Rde (data is shown in Fig. 1B,C) | Rde (data is shown in Fig. 1B,C) |
| *gk925207* | R1165K | S | benign | TOLERATED | neutral | 100 ± 0 | 100 ± 0 |
| *gk749674* | G1188E | G (conserved) | probably damaging | DELETERIOUS | Deleterious | 98 ± 2 | 96.3 ± 3.2 |
| *gk532049* | S1198L | S (conserved) | possibly damaging | TOLERATED | Deleterious | Rde (data is shown in Fig. 1B,C) | Rde (data is shown in Fig. 1B,C) |
| *gk115391* | E1278K | E (conserved) | benign | TOLERATED | neutral | 100 ± 0 | 100 ± 0 |
| *gk317493* | E1397K | I | benign | NA | neutral | 99.7 ± 0.6 | 100 ± 0 |
| *gk115390* | K1469E | K (conserved) | possibly damaging | NA | neutral | 100 ± 0 | 100 ± 0 |
| *gk864727* | N1577S | T | benign | NA | neutral | 100 ± 0 | 100 ± 0 |
| *gk540555* | G1610R | G (conserved) | probably damaging | NA | neutral | 100 ± 0 | 100 ± 0 |

All RNAi data represent the values of three technical replicates.

## Expression and subcellular localization of EGO-1(S1198L) are comparable to the wildtype form EGO-1

To study the endogenous protein localization of EGO-1(S1198L), we tagged the EGO-1(S1198L) protein by CRISPR/Cas9-mediated insertion of GFP at its N-terminus (see Materials and Methods). GFP::EGO-1(S1198L) was detected as punctate structures associated with P granules around the nucleus, similar to the wildtype form of GFP::EGO-1 (Chen et al, 2024) (Fig. EV3A). The fluorescence intensity showed no significant difference between the wildtype form and S1198L (Fig. EV3B), suggesting that the EGO-1(S1198L) mutation induces RNAi defects without causing abnormalities in subcellular localization or expression levels.

## Loss of ZNFX-1 partially suppresses the germline exo-RNAi defects of *ego-1(S1198L)*

Previous studies have implicated EGO-1 in P granule biogenesis (Vought et al, 2005). P granules serve as safe harbors where germline-expressed mRNA is protected from silencing by PRG-1 and HRDE-1 (Dodson and Kennedy, 2019; Lev et al, 2019; Ouyang et al, 2019). Additionally, it has been shown that EGO-1 binds to ZNFX-1, a Z granule component required for RNAi inheritance, together with HRDE-1 (Ishidate et al, 2018; Ouyang et al, 2022). To determine whether these granules were affected, we analyzed the patterns of PGL-1::tagRFP and GFP::ZNFX-1 in the germline. Perinuclear ZNFX-1 and PGL-1 puncta were observed in the wild-type pachytene region. The overall pattern of these puncta was not affected in *ego-1(S1198L)* homozygotes (Fig. EV4A,B). However, in *ego-1(S1198L)* homozygotes, the fluorescence intensity per punctum significantly increased for both PGL-1::tagRFP and GFP::ZNFX-1 (Fig. EV4C,D). Since the fluorescence intensity per punctum for GFP::ZNFX-1 increased in *ego-1(S1198L)* homozygotes, we next examined germline exo-RNAi activity in double mutants for *ego-1(S1198L)* and *znfx-1(gg561)*. *znfx-1(gg561)* suppressed the *pos-1*, *gld-1* and *mpk-1* RNAi defects in *ego-1(S1198L)* (Figs. 3A,B and EV5A). Although not statistically significant, *znfx-1(gg561)* slightly increased sensitivity to *pop-1*

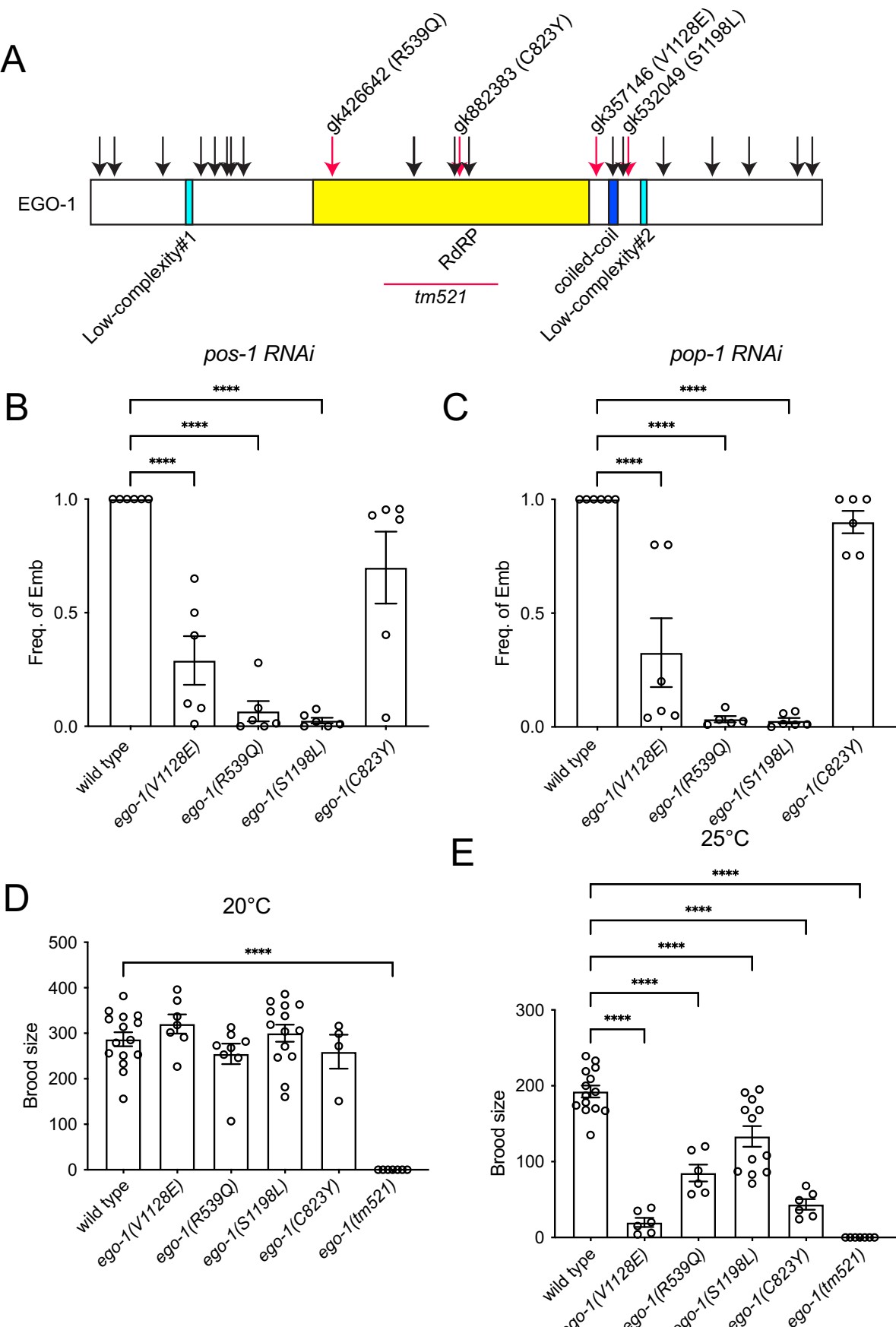

◀ **Figure 1. The non-null mutations of *ego-1*, which cause germline exo-RNAi defects.**

(A) Protein structure of EGO-1 showing mutations that were used in this study. The mutations that cause germline exo-RNAi defects are indicated by red arrows, while those that do not cause such defects are marked with black arrows. (B) Quantification of the frequency of indicated mutant animals showing the Emb phenotype under *pos-1* feeding RNAi. Wildtype and the *gk357146 (V1128E)*, *gk426642 (R539Q)*, *gk532049 (S1198L)*, and *gk882383 (C823Y)* mutants were analyzed (n = 6 technical replicates for each genotype). Error bars represent mean ± SEM. *P* values were determined using one-way ANOVA with Dunnett's multiple-comparisons test compared to wildtype. **** indicates *p* value <0.001. *p* < 0.0001 for V1128E, R539Q, and S1198L. For C823Y, *p* = 0.0737. (C) Quantification of the frequency of indicated mutant animals showing the Emb phenotype under *pop-1* feeding RNAi. Wildtype (n = 6), *gk357146 (V1128E)* (n = 6), *gk426642 (R539Q)* (n = 5), *gk532049 (S1198L)* (n = 6), and *gk882383 (C823Y)* (n = 6) were examined, with n indicating the number of technical replicates. Error bars represent mean ± SEM. *P* values were determined using one-way ANOVA with Dunnett's multiple-comparisons test compared to wildtype. **** indicates *p* value <0.001. *p* < 0.0001 for V1128E, R539Q, and S1198L. For C823Y, *p* = 0.7426. (D, E) Brood size analysis of the indicated mutant animals that were cultured at 20 °C (D) and 25 °C (E). (D) Wildtype (n = 15), *ego-1(tm521)* (n = 7), *ego-1(V1128E)* (n = 7), *ego-1(R539Q)* (n = 8), *ego-1(S1198L)* (n = 14), *ego-1(C823Y)* (n = 4), and (E) wildtype (n = 14), *ego-1(tm521)* (n = 7), *ego-1(V1128E)* (n = 6), *ego-1(R539Q)* (n = 6), *ego-1(S1198L)* (n = 12) and *ego-1(C823Y)* (n = 6) were examined, with n indicating the number of biological replicates. *P* values were determined using one-way ANOVA with Dunnett's multiple-comparisons test compared to wildtype. **** indicates *p* value <0.001. (D) *p* < 0.0001 for *ego-1(tm521)*. *P* values for other comparisons were: V1128E, *p* = 0.6743; R539Q, *p* = 0.6806; S1198L, *p* = 0.9729; C823Y, *p* = 0.9137. (E) For all comparisons shown, *p* < 0.0001. Source data are available online for this figure.

RNAi (Fig. EV5B). These results suggest that the *ego-1(S1198L)* mutation upregulates the function of ZNFX-1 and/or Z granules.

## Loss of HRDE-1 partially suppresses the germline exo-RNAi defects in *ego-1(S1198L)*

HRDE-1 is a nuclear Argonaute protein that mediates nuclear RNAi and inheritance of piRNA silencing in germ cells (Ashe et al, 2012; Buckley et al, 2012; Luteijn et al, 2012; Shirayama et al, 2012). HRDE-1-dependent gene silencing has been shown to inhibit the RNA polymerase II-dependent RNA transcription of germline genes, including *sid-1* and *rde-11*, which are required for exo-RNAi (Dodson and Kennedy, 2019; Ouyang et al, 2019). To test whether the Rde phenotype was caused by the HRDE-1-mediated silencing of germline exo-RNAi genes, we examined the genetic interactions between *hrde-1(tm1200)* and *ego-1(S1198L)*. The *hrde-1* single mutants showed normal responses to *pos-1* and *pop-1* RNAi, as reported previously (Yigit et al, 2006), while they were slightly resistant to *gld-1* and *mpk-1* RNAi (Figs. 3A and EV5). Similar to *znfx-1(gg561)*, *hrde-1(tm1200)* partially suppressed the Rde phenotype of *pos-1*, *gld-1* and *mpk-1* RNAi in *ego-1(S1198L)* (Figs. 3A and EV5A).

## HRDE-1::GFP accumulates in *ego-1(S1198L)* pachytene-stage cells

Since germline exo-RNAi defects in *ego-1(S1198L)* were suppressed in the absence of HRDE-1, we wondered whether the distribution of the HRDE-1 protein was altered in *ego-1(S1198L)*. GFP::HRDE-1 was predominantly localized to the nuclei in wild-type pachytene-stage cells, as reported previously (Fig. 3C) (Ashe et al, 2012; Buckley et al, 2012). In *ego-1(S1198L)* homozygous pachytene-stage cells, both nuclear and cytosolic levels of HRDE-1::GFP increased, and germ granule-like dots were detected (Fig. 3C,D). Additionally, the nuclear/cytosolic ratio decreased slightly (Fig. 3E). The overall accumulation of the HRDE-1 protein in the *ego-1(S1198L)* mutant suggests that EGO-1 activity is required, directly or indirectly, for maintaining the normal homeostasis of HRDE-1 protein levels. The precise mechanism linking EGO-1's RdRP function to HRDE-1 protein regulation remains to be determined.

## The expression of *sid-1* and *rde-11* is downregulated in *ego-1(S1198L)* in an HRDE-1-dependent manner

To evaluate the expression of *sid-1* and *rde-11* in the mutants, RT-qPCR was performed. Expression of these genes was

significantly downregulated in the *ego-1(S1198L)* mutant (Fig. 4A,B). Similarly, expression of these genes was significantly downregulated in the *ego-1(tm521)* mutant (Fig. 4C,D). The reduced levels observed in *ego-1(S1198L)* and *ego-1(tm521)* were comparable, suggesting that the *ego-1(S1198L)* mutation-induced defects are equivalent to loss of function in terms of the regulation of gene expression. The expression levels of *sid-1* and *rde-11* were significantly increased in the double mutants of *ego-1(S1198L)* and *hrde-1(tm1200)* (Fig. 4A,B). Notably, the expression of *rde-11* in these double mutants was significantly restored to wild-type levels (Fig. 4B). These data suggest that EGO-1 affects the regulation of 22G RNA loading on HRDE-1, which in turn contributes to the germline expression of *sid-1* and *rde-11*.

## Characterization of germline exo-RNAi phenotypes in *ego-1* mutant heterozygotes

RRF-1 compensates for partial loss of EGO-1 activity in S1198L with respect to 25 °C brood size (Fig. 2C), but not for germline exo-RNAi (Fig. EV2C,D). Therefore, the defects of *ego-1(S1198L)* in germline exo-RNAi could manifest as inhibition of an exo-RNAi process that requires both EGO-1 and RRF-1. In this scenario, *ego-1(S1198L)* should be genetically dominant among germline exo-RNAi. To address this, we examined the germline exo-RNAi sensitivity of *ego-1(S1198L)* heterozygous hermaphrodites. Consistent with this possibility, the heterozygous hermaphrodites for the *ego-1(S1198L)*, derived from homozygous hermaphrodites (Fig. EV6A,B), and wild-type (*tmC18[tmIs1200]/+*) males displayed the Rde phenotype. In contrast, heterozygotes for the *ego-1* deletion mutant allele, *ego-1(tm521)*, showed a normal germline exo-RNAi response (Fig. EV6A,B). This dominant effect was also observed in *ego-1(R539Q)*, but not in *ego-1(V1128E)* or *ego-1(C823Y)* heterozygotes (Fig. EV6D,E). We also found that *ego-1(S1198L)/ego-1(tm521)* trans-heterozygotes, derived from *ego-1(S1198L)* homozygous hermaphrodites and *ego-1(tm521)* heterozygous males, exhibited a slight reduction in brood size at 25 °C, comparable to that of *ego-1(S1198L)* heterozygotes (Fig. EV6C). In addition, they displayed the Rde phenotype when subjected to *pos-1* and *pop-1* RNAi (Fig. EV6D,E). These data, along with the results of the following experiments, suggest that the observed phenotypes of *ego-1(S1198L)* heterozygotes thus far resulted from dominant effects—particularly epigenetic mechanisms (see below) and possibly antimorphic effects—rather than haploinsufficiency at the *ego-1* locus.

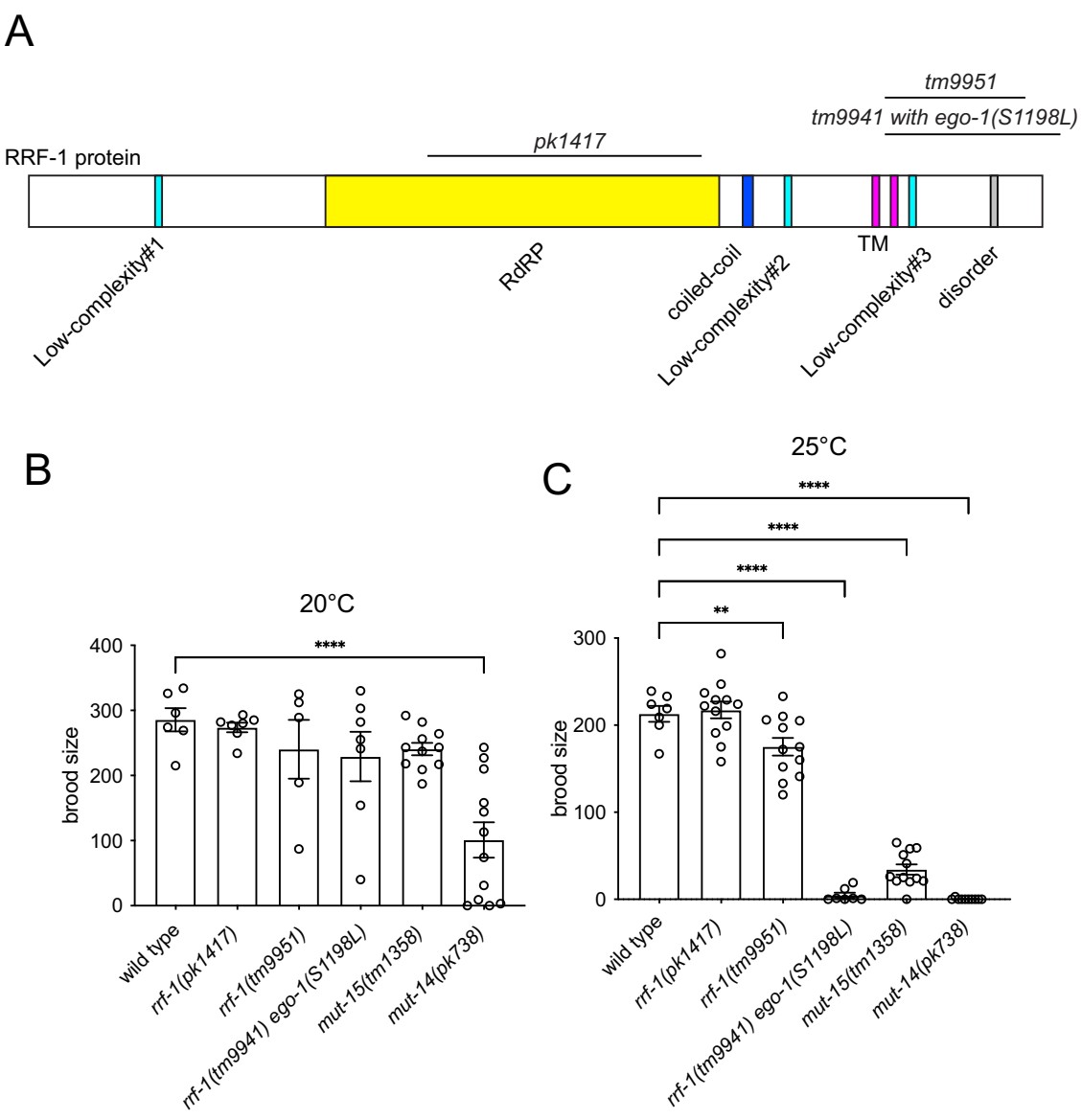

**Figure 2. Combining the *ego-1(S1198L)* mutation with the *rrf-1* deletion mutation results in a temperature-sensitive sterile phenotype.**

(A) Protein structure of RRF-1 showing the mutations used in this study. TM: transmembrane domain. (B, C) Brood size analysis of the indicated mutant animals that were cultured at 20 °C (B) and 25 °C (C). (B) Wildtype (*n* = 6), *rrf-1(pk1417)* (*n* = 7), *rrf-1(tm9951)* (*n* = 5), *rrf-1(tm9941) ego-1(S1198L)* (*n* = 7), *mut-15(tm1358)* (*n* = 11), and *mut-14(pk738)* (*n* = 12) and (C) wildtype (*n* = 7), *rrf-1(pk1417)* (*n* = 12), *rrf-1(tm9951)* (*n* = 12), *rrf-1(tm9941) ego-1(S1198L)* (*n* = 7), *mut-15(tm1358)* (*n* = 12), and *mut-14(pk738)* (*n* = 11) were examined, with n indicating the number of biological replicates. *P* values were determined using one-way ANOVA with Dunnett's multiple-comparisons test compared to wildtype. ** indicates *p* value <0.01, and **** indicates *p* value <0.001. (B) *p* < 0.0001 for *mut-14(pk738)*. *P* values for other comparisons were: *rrf-1(pk1417)*, *p* = 0.9979; *rrf-1(tm9951)*, *p* = 0.7272; *rrf-1(tm9941) ego-1(S1198L)*, *p* = 0.4726; *mut-15(tm1358)*, *p* = 0.5931. (C) *P* values for other comparisons were: *rrf-1(pk1417)*, *p* = 0.9979; *rrf-1(tm9951)*, *p* = 0.7272; *rrf-1(tm9941) ego-1(S1198L)*, *p* = 0.4726; *mut-15(tm1358)*, *p* = 0.5931. Source data are available online for this figure.

## Disconnection between the *ego-1* genotype and phenotype across generations

Disruption of P granule formation results in aberrant siRNA expression and abnormal suppression of germline RNAi genes, including *sid-1* and *rde-11*, over several generations (Dodson and Kennedy, 2019; Ouyang et al, 2019). To determine whether this phenomenon underlies the germline Rde phenotype observed in *ego-1(S1198L)* heterozygotes, we investigated the germline exo-RNAi-defective phenotype over generations following mating with

*tmC18[tmIs1200]*, which is a wild-type for the *ego-1* gene. We found that *ego-1(+)* homozygotes (genotype *tmC18[tmIs1200]/tmC18[tmIs1200]*), which originated from *ego-1(S1198L)* heterozygous hermaphrodites derived from *ego-1(S1198L)* homozygous hermaphrodites and wild-type (*tmC18[tmIs1200]/+*) males, exhibited a germline exo-RNAi defective phenotype (Fig. 5A–C). We also found that the germline RNAi-defective phenotype of *ego-1(+)*, originating from *ego-1(S1198L)* heterozygous hermaphrodites, persisted for five generations (Fig. 5E,F), indicating that the germline exo-RNAi defects found in *ego-1(S1198L)/*

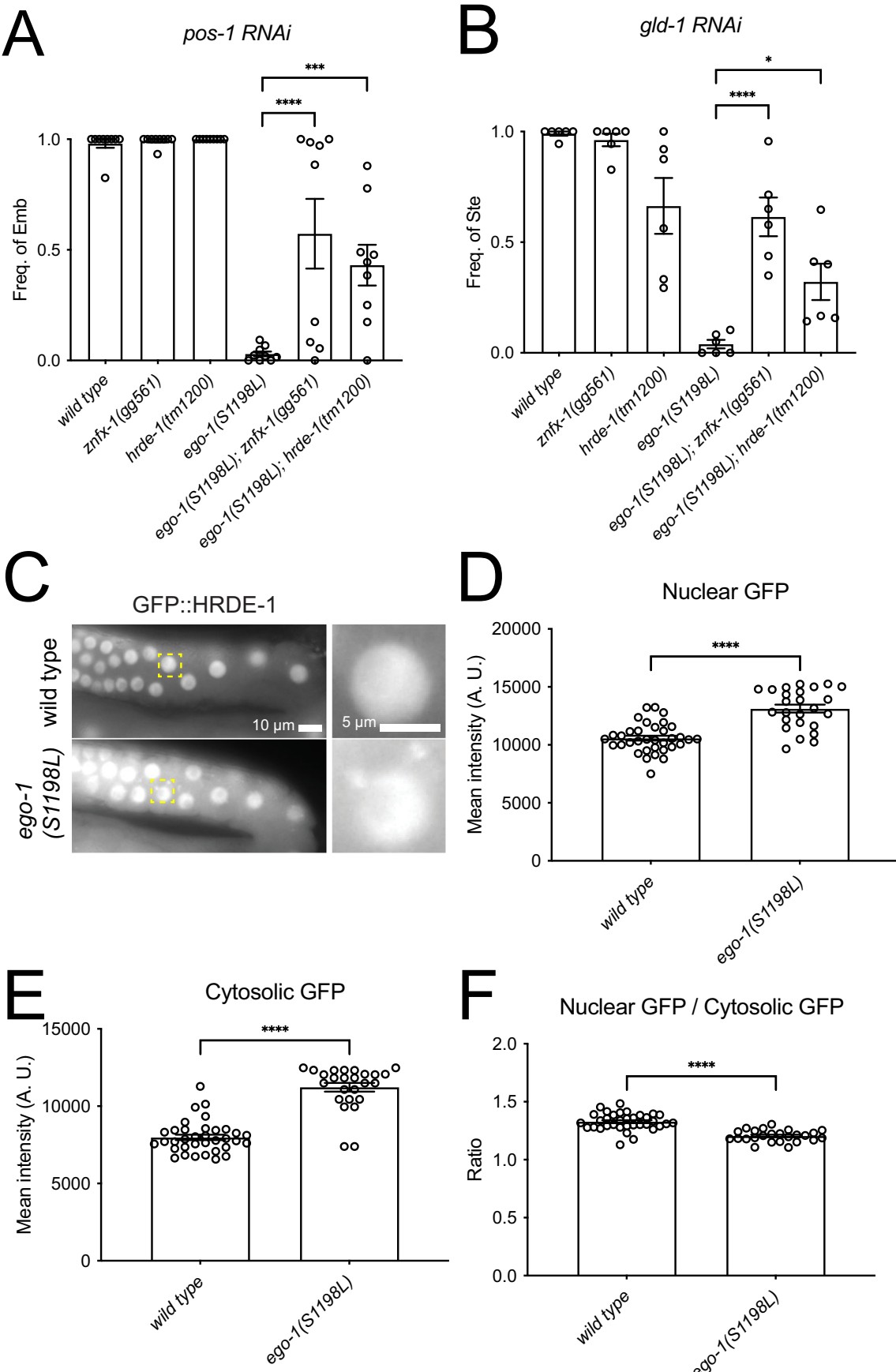

**Figure 3. Germline exo-RNAi defects in *ego-1(S1198L)* are partially suppressed by *hrde-1* and *znfx-1* mutations.**

(A) Quantification of the frequency of indicated mutant animals showing the Emb phenotype under *pos-1* feeding RNAi. $n = 9$ technical replicates for each genotype were examined. Error bars represent mean ± SEM. *P* values were determined using one-way ANOVA with Šídák's multiple-comparisons test, compared to the *ego-1(S1198L)*. *** indicates $p < 0.001$ and **** indicates $p < 0.0001$. For the comparison with *ego-1(S1198L); znfx-1(gg561)*, $p < 0.0001$; for *ego-1(S1198L); hrde-1(tm1200)*, $p = 0.0009$. (B) Quantification of the frequency of indicated mutant animals showing the Ste phenotype under *gld-1* feeding RNAi. $n = 6$ technical replicates for each genotype were examined. Error bars represent mean ± SEM. *P* values were determined using one-way ANOVA with Šídák's multiple-comparisons test, compared to the *ego-1(S1198L)*. * indicates $p < 0.05$ and **** indicates $p > 0.0001$. For the comparison with *ego-1(S1198L); hrde-1(tm1200)*, $p = 0.0204$; for *ego-1(S1198L); znfx-1(gg561)*, $p < 0.0001$. (C) Fluorescence images of germ cells showing the localization of GFP::HRDE-1. (D) Bar graph showing the quantification of nuclear GFP::HRDE-1 fluorescence signal intensity. Error bars represent mean ± SEM. (E) Bar graph showing the quantification of cytoplasmic GFP::HRDE-1 fluorescence signal intensity. Error bars represent mean ± SEM. (F) Bar graph showing the ratio of nuclear GFP/cytoplasmic GFP signals. Error bars represent mean ± SEM. Data in (D–F) are derived from the same image set, where wildtype ($n = 35$, one nucleus per animal) and *ego-1(S1198L)* mutants ($n = 25$, one nucleus per animal) were examined. Statistical significance was determined using two-tailed unpaired *t*-tests. In all cases, the difference was significant (****$p < 0.0001$). Source data are available online for this figure.

*tmC18[tmIs1200]* heterozygotes resulted from epigenetic inheritance initiated by the homozygous *ego-1(S1198L)* mutation. Interestingly, *ego-1(S1198L)/tmC18[tmIs1200]* heterozygotes exhibited more persistent RNAi defects than *ego-1(+)* originating from *ego-1(S1198L)* heterozygous hermaphrodites, suggesting that *ego-1(S1198L)* likely functions predominantly in this context (Fig. 5D–F). Notably, such a persistence of RNAi defects in *ego-1(+)* homozygotes and heterozygous animals was not observed in another exo-RNAi-defective mutant, *ego-1(V1128E)*, suggesting that this phenotype is specific to the S1198L mutation (Fig. EV7).

Given the redundant involvement of EGO-1 and RRF-1 in temperature-dependent sterility, we examined germline exo-RNAi activity in the descendants of *rrf-1(tm9941) ego-1(S1198L)* double-mutant animals. In contrast to *ego-1(+)* animals from *ego-1(S1198L)*, this epigenetic effect was not observed in *rrf-1(+) ego-1(+)* animals from *rrf-1(tm9941) ego-1(S1198L)*, suggesting that ancestral RRF-1 is involved in the establishment of this epigenetic state (Fig. 5B,C). Taken together, these findings highlight the counteractive interplay of RdRPs in the inheritance of germline exo-RNAi-defective phenotypes, along with the redundant enzymatic activity directly required for the silencing of exogenous genes.

## RRF-1 is required for the persistent silencing of *sid-1* and *rde-11* over generations in the descendants of *ego-1(S1198L)*

The *rrf-1(tm9941) ego-1(S1198L)* double mutant itself exhibited germline exo-RNAi defects, but these defects did not persist in the wild-type animals derived from this strain. To investigate how RRF-1 affects the regulation of germline exo-RNAi genes, we analyzed the expression of *sid-1* and *rde-11* in *ego-1(S1198L)* and *rrf-1(tm9951)* single mutants and *rrf-1(tm9941) ego-1(S1198L)* double-homozygous mutants cultured for more than ten generations. The levels of *sid-1* and *rde-11* expression in *rrf-1(tm9951)* animals were similar to those in wild-type animals (Fig. 6A,B). Unexpectedly, in *rrf-1(tm9941) ego-1(S1198L)* double-mutant animals, the levels of *sid-1* and *rde-11* were significantly decreased compared to those in the wild-type, but were comparable to those in *ego-1(S1198L)* single-mutant animals (Fig. 6A,B). Although *rrf-1(tm9941)* does not restore *sid-1* and *rde-11* expression when combined with *ego-1(S1198L)* (Fig. 6A,B), the ancestral *rrf-1(tm9941) ego-1(S1198L)* facilitates higher *sid-1* and *rde-11* expression in their *rrf-1(+) ego-1(+)* G3-G5 descendants compared to ancestral *ego-1(S1198L)* (see below, Fig. 6C,D). The molecular mechanisms behind these differences are not clear. These findings

indicate that while RRF-1 is involved in the persistent silencing of *sid-1* and *rde-11*, its absence does not necessarily reverse the expression defects caused by *ego-1(S1198L)*. Next, we examined the expression of *sid-1* and *rde-11* in wild-type descendants derived from *ego-1(S1198L)* and *rrf-1(tm9941) ego-1(S1198L)* mutant animals at generations 3, 5, 7, and 10 after crossing with the *tmC18* balancer (Fig. 6C,D). In the descendants, the expression level of *sid-1* recovered progressively with each generation when derived from *ego-1(S1198L)* mutants, whereas it recovered in the *rrf-1(tm9941) ego-1(S1198L)* mutant in the early generations (Fig. 6C). The expression level of *rde-11* in *ego-1(S1198L) rrf-1(tm9941)* mutant-derived descendants was higher than that in *ego-1(S1198L)* mutant-derived descendants (Fig. 6D). These results suggest that EGO-1(S1198L) is defective in licensing the expression of *sid-1* and *rde-11*. In contrast, RRF-1 mediated the silencing of *sid-1* and *rde-11* when EGO-1 function was inhibited.

## R539Q shows the transgenerational RNAi defective phenotype and the reduction of *rde-11* expression

Notably, the *ego-1(S1198L)* mutants are fertile, unlike the null allele *ego-1(tm521)*, which is sterile. This feature may uniquely permit the inheritance of RNAi-resistant states over generations. Thus, the special nature of the S1198L allele likely derives from its dual characteristics: a functional alteration that impairs germline exo-RNAi, and sufficient preservation of germline function to allow progeny production and facilitate transgenerational analysis. Given this unique combination of phenotypes, we then examined whether the RNAi defective phenotype across generations, as well as the reduction of *rde-11* and *sid-1* expression, observed in *ego-1(S1198L)*, are uniquely associated with this mutation by analyzing other fertile *ego-1* MMP mutants. Among the *ego-1* MMP mutants tested, only the *ego-1(+)* homozygotes derived from the R539Q homozygous strain, which exhibits strong germline exo-RNAi defects (Fig. 1B,C), showed a significant RNAi defective phenotype across generations (Table 2). Thus, one possible interpretation is that the "Rde hangover" phenotype may result from a moderately severe dysfunction of EGO-1 that does not affect fertility, but disrupts the coordination of sRNA-mediated regulation across generations. Notably, *rde-11* expression was significantly reduced in this strain, whereas *sid-1* levels remained unchanged (Table 3). This contrasts with the S1198L mutant, in which both *rde-11* and *sid-1* expression are reduced (Fig. 4A,B). These findings indicate that the transcriptional status of *sid-1* alone cannot fully explain the occurrence of the transgenerational phenotype. Supporting this,

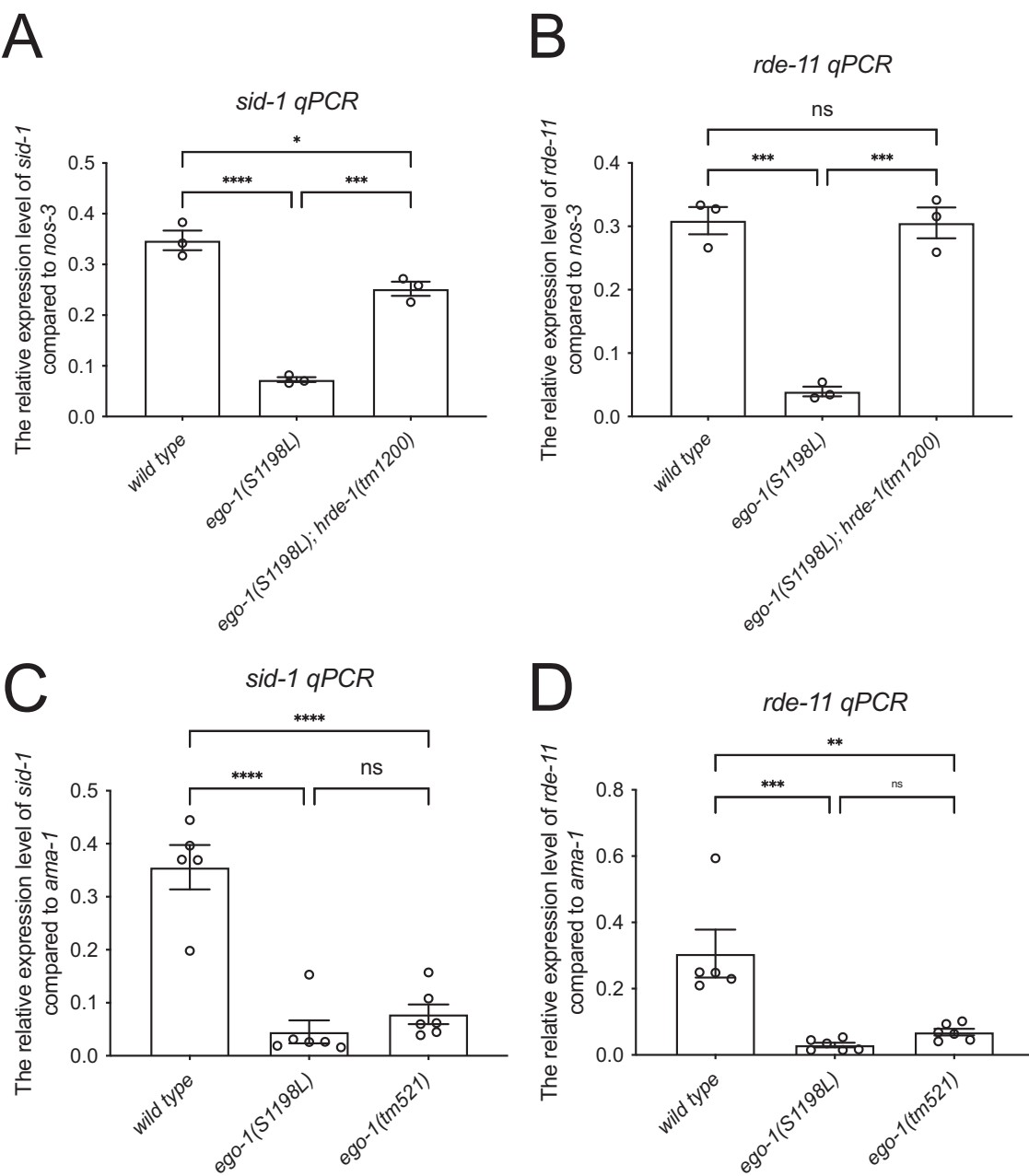

**Figure 4. ego-1(S1198L) shows reduced expression of sid-1 and rde-11 in an HRDE-1-dependent manner.**

(A, B) RT-qPCR showing the relative expression levels of sid-1 (A) and rde-11 (B) mRNA normalized to nos-3 expression. RNA was extracted from young-adult animals grown at 20 °C. Data represent values from three biological replicates. Error bars represent mean ± SEM. (C, D) RT-qPCR showing the relative expression levels of sid-1 (C) and rde-11 (D) mRNA normalized to ama-1 expression. Data represent values from five biological replicates (wildtype) or six biological replicates (other genotypes). Error bars represent mean ± SEM. RNA was extracted from young-adult animals grown at 20 °C. Statistical analysis was performed using one-way ANOVA with Šídák's multiple-comparisons test. (A) * indicates $p < 0.05$, *** indicates $p < 0.001$, and **** indicates $p < 0.0001$. P values for specific comparisons were as follows: wildtype vs. ego-1(S1198L), $p < 0.0001$; wildtype vs. ego-1(S1198L); hrde-1(tm1200), $p = 0.0116$; and ego-1(S1198L) vs. ego-1(S1198L); hrde-1(tm1200), $p = 0.0004$. (B) *** indicates $p < 0.001$. P values for specific comparisons were as follows: ego-1(S1198L) vs. ego-1(S1198L); hrde-1(tm1200), $p = 0.0003$; wildtype vs. ego-1(S1198L); hrde-1(tm1200), $p = 0.9999$; and wildtype vs. ego-1(S1198L), $p = 0.0002$. (C) **** indicates $p < 0.0001$. P-values for specific comparisons were as follows: wildtype vs. ego-1(S1198L), $p < 0.0001$; wildtype vs. ego-1(tm521), $p < 0.0001$; and ego-1(S1198L) vs. ego-1(tm521), $p = 0.7752$. (D) ** indicates $p < 0.01$ and *** indicates $p < 0.001$. P values for specific comparisons were as follows: wildtype vs. ego-1(S1198L), $p = 0.0004$; wildtype vs. ego-1(tm521), $p = 0.0017$; and ego-1(S1198L) vs. ego-1(tm521), $p = 0.8408$. Source data are available online for this figure.

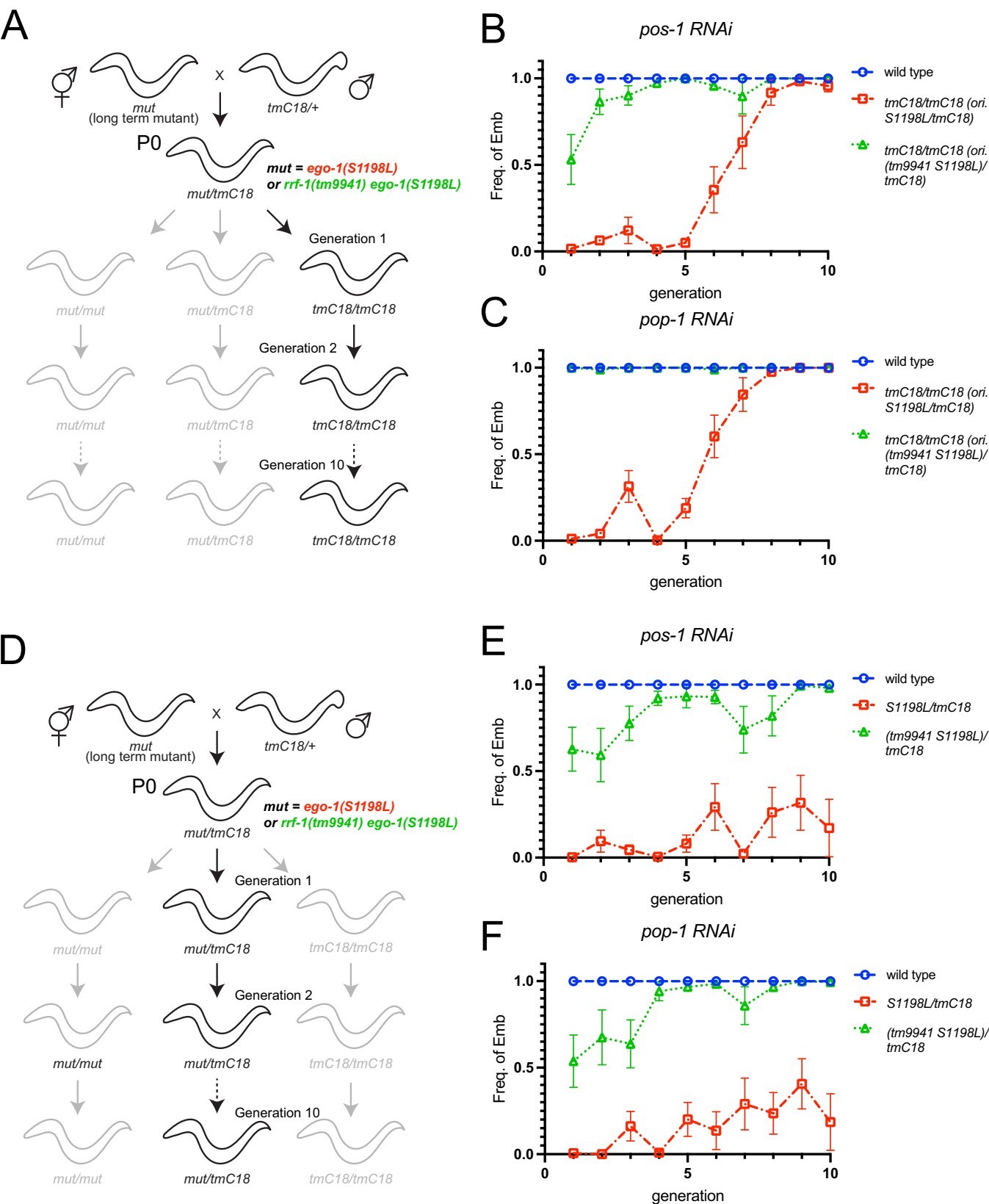

◀ **Figure 5. Transgenerational effects of *ego-1(S1198L)* on germline exo-RNAi.**

(A, D) Schematics of genetic crosses. P0 is the first generation in which cross-progenies are selected. The *tmC18* used in this experiment harbors *tmIs1200[dpy-5(Pmyo-2::Venus)]* in the region of the rearranged chromosome, which can stably maintain the *ego-1* and *rrf-1* mutations in trans-heterozygotes. Therefore, the copy number of the *ego-1* single and *ego-1 rrf-1* double mutations can be determined based on the phenotype and fluorescence expression caused by the markers contained in *tmC18*. This means that in progeny arising from parents balanced by *tmC18*, Dpy worms expressing GFP in the pharynx are those without the *ego-1* or *ego-1 rrf-1* mutations, non-Dpy worms expressing GFP are heterozygous for the *ego-1* or *ego-1 rrf-1* mutations, and non-Dpy worms not expressing GFP are homozygous for the *ego-1* or *ego-1 rrf-1* mutations. (B, C) Quantification of the frequency of the indicated mutant animals showing the Emb phenotype under *pos-1* (B) and *pop-1* (C) feeding RNAi in the indicated generation. Error bars represent mean ± SEM. The wild-type was represented by N2 animals. *tmC18/tmC18* (ori. *S1198L/tmC18*) and *tmC18/tmC18* (ori. *tm9941 S1198L/tmC18*) are descendants in A, where mut represents *ego-1(S1198L)* or *rrf-1(tm9941) ego-1(S1198L)*. (B) For wildtype and *tmC18/tmC18* (ori. *gk532049/tmC18*), $n = 9$ biological replicates were obtained for generations 1–9, and $n = 6$ for generation 10. For *tmC18/tmC18* (ori. *(tm9941 gk532049)/tmC18*), $n = 9$ biological replicates were obtained for generations 1–5, 7, 8, $n = 8$ for generation 6, and $n = 6$ for generations 9 and 10. (C) For wildtype, $n = 9$ biological replicates were obtained for generations 1–7, and $n = 6$ for generations 8–10. For *tmC18/tmC18* (ori. *gk532049/tmC18*), $n = 9$ biological replicates were obtained for generations 1–3, 5, and 6, $n = 7$ for generations 4 and 7, and $n = 6$ for generations 8–10. For *tmC18/tmC18* (ori. *(tm9941 gk532049)/tmC18*), $n = 9$ biological replicates were obtained for generations 1–3 and 5–7, $n = 8$ for generation 4, $n = 6$ for generations 9 and 10, and $n = 5$ for generation 8. (E, F) Quantification of the frequency of the indicated mutant animals showing the Emb phenotype under *pos-1* (E) and *pop-1* (F) feeding RNAi in the indicated generation. Error bars represent mean ± SEM. The wild-type was represented by N2 animals. *S1198L/tmC18* and *(tm9941 S1198L)/tmC18* are the descendants indicated in (D), where mut represents *ego-1(S1198L)* or *rrf-1(tm9941) ego-1(S1198L)*. (E) For wildtype, $n = 9$ biological replicates for generations 1–9, and $n = 7$ for generation 10. For *tmC18/tmC18* (ori. *gk532049/tmC18*), $n = 9$ biological replicates for generations 1–9, and $n = 8$ for generation 10. For *tmC18/tmC18* (ori. *(tm9941 gk532049)/tmC18*), $n = 9$ biological replicates for generations 1–5, 7, and 8, $n = 8$ for generation 6, and $n = 7$ for generations 9 and 10. (F) For wildtype, $n = 9$ biological replicates for generations 1–9, and $n = 6$ for generation 10. For *tmC18/tmC18* (ori. *gk532049/tmC18*), $n = 9$ biological replicates for generations 1–9, and $n = 6$ for generation 10. For *tmC18/tmC18* (ori. *(tm9941 gk532049)/tmC18*), $n = 9$ biological replicates for generations 1–9, and $n = 6$ for generations 9 and 10. Source data are available online for this figure.

previous studies demonstrated that in *meg-3/4* mutants, siRNA accumulation is altered not only for *rde-11* and *sid-1*, but also for additional genes involved in chromatin regulation and RNAi amplification (*hda-3*, *zfp-1*, *set-23*, *wago-2*) (Dodson and Kennedy, 2019; Ouyang et al, 2019). It is therefore plausible that the penetrance and persistence of the "Rde hangover" phenotype arise from a combinatorial network of direct and indirect regulatory effects involving these multiple factors, particularly in mutants that exhibit strong germline RNAi resistance in the homozygous state.

## Discussion

Investigating the unique roles of EGO-1 and RRF-1 is valuable, as it helps explain gene expression regulation through different RNAi mechanisms, including those initiated by exogenous dsRNA. Although it has been suggested that these molecules may have overlapping functions in exo-RNAi (Sijen et al, 2001; Smardon et al, 2000), the underlying mechanisms remain unclear. Previous studies have reported that P granules protect *sid-1* and *rde-11* mRNAs from PRG-1/HRDE-1-dependent silencing (Dodson and Kennedy, 2019; Lev et al, 2019; Ouyang et al, 2019). Additionally, CSR-1 protects its targets from silencing by PRG-1(Seth et al, 2013; Wedeles et al, 2013). Given that EGO-1 and RRF-1 are responsible for the synthesis of CSR-1-class and WAGO-class 22G RNAs, respectively, a model can be considered in which RRF-1 amplifies 22G RNAs in PRG-1/HRDE-1-dependent silencing, whereas EGO-1 protects targets from PRG-1/HRDE-1-mediated silencing (Fig. 7A–D). However, testing this hypothesis is challenging because *ego-1* null mutants are sterile. In this study, we provide experimental evidence supporting this hypothesis. We analyzed *ego-1* alleles in the MMP collection, which revealed germline exo-RNAi defects in *ego-1(S1198L)*. Despite exhibiting these defects, this allele retains its fundamental function in temperature-independent germline development, providing valuable insights into the specific role of EGO-1 in RNAi mechanisms (Fig. 7A–D). Additionally, this allele showed synthetic temperature-sensitive sterility with an *rrf-1* deletion mutation. Germline exo-RNAi

defects in *ego-1(S1198L)* were partially suppressed by the inhibition of HRDE-1 and ZNFX-1, whereas inheritance of the RNAi-defective phenotype across generations occurred in an RRF-1-dependent manner. The transgenerational inheritance of the RNAi-defective phenotype observed in *ego-1(S1198L)* was previously described as a "phenotypic hangover" or "Rde hangover phenotype", where the RNAi-defective phenotype persists even in the absence of the original gene mutation (Dodson and Kennedy, 2019; Lev et al, 2019; Ouyang et al, 2019). The Rde hangover phenotype is likely triggered by an ancestral loss of the organizing capabilities of P granules that play a role in protection of germline mRNA, such as *sid-1* and *rde-11*, from HRDE-1-dependent germline gene silencing (Dodson and Kennedy, 2019; Lev et al, 2019; Ouyang et al, 2019). Our study demonstrates that, in this context, although EGO-1 functions redundantly with RRF-1 to undergo temperature-dependent gametogenesis, EGO-1 and RRF-1 counteract each other during HRDE-1-dependent germline gene silencing. Thus, the dual role of EGO-1 highlights its importance in enhancing germline robustness to exo-RNAi via RdRP activity, which is directly required for target gene silencing and protecting exo-RNAi genes from HRDE-1-dependent silencing.

Notably, strong RNAi-defective alleles (S1198L and R539Q) do not exhibit sterility at high temperatures, whereas weaker alleles (C823Y and V1128E) show pronounced fertility defects. This suggests that sterility may be influenced by distinct functional properties of these mutations. One possible explanation for the weaker impact of S1198L on fertility is that this mutation confers unique effects beyond partial loss of function. This mutation may not only impair EGO-1 activity but also enhance RRF-1 function. Supporting this idea, S1198L heterozygotes exhibit transgenerational persistence of germline RNAi defects, suggesting a potential upregulation of RRF-1 activity. Future studies investigating RRF-1 enzymatic activity and subcellular localization across *ego-1* mutants will provide deeper insights into their molecular interactions.

We found that *znfx-1* and *hrde-1* mutations partially restored the sensitivity of *ego-1(S1198L)* mutants to RNAi targeting *pos-1*, *gld-1* and *mpk-1*, but did not significantly restore *pop-1* RNAi. The variation in exogenous RNAi (exo-RNAi) efficiency caused by the

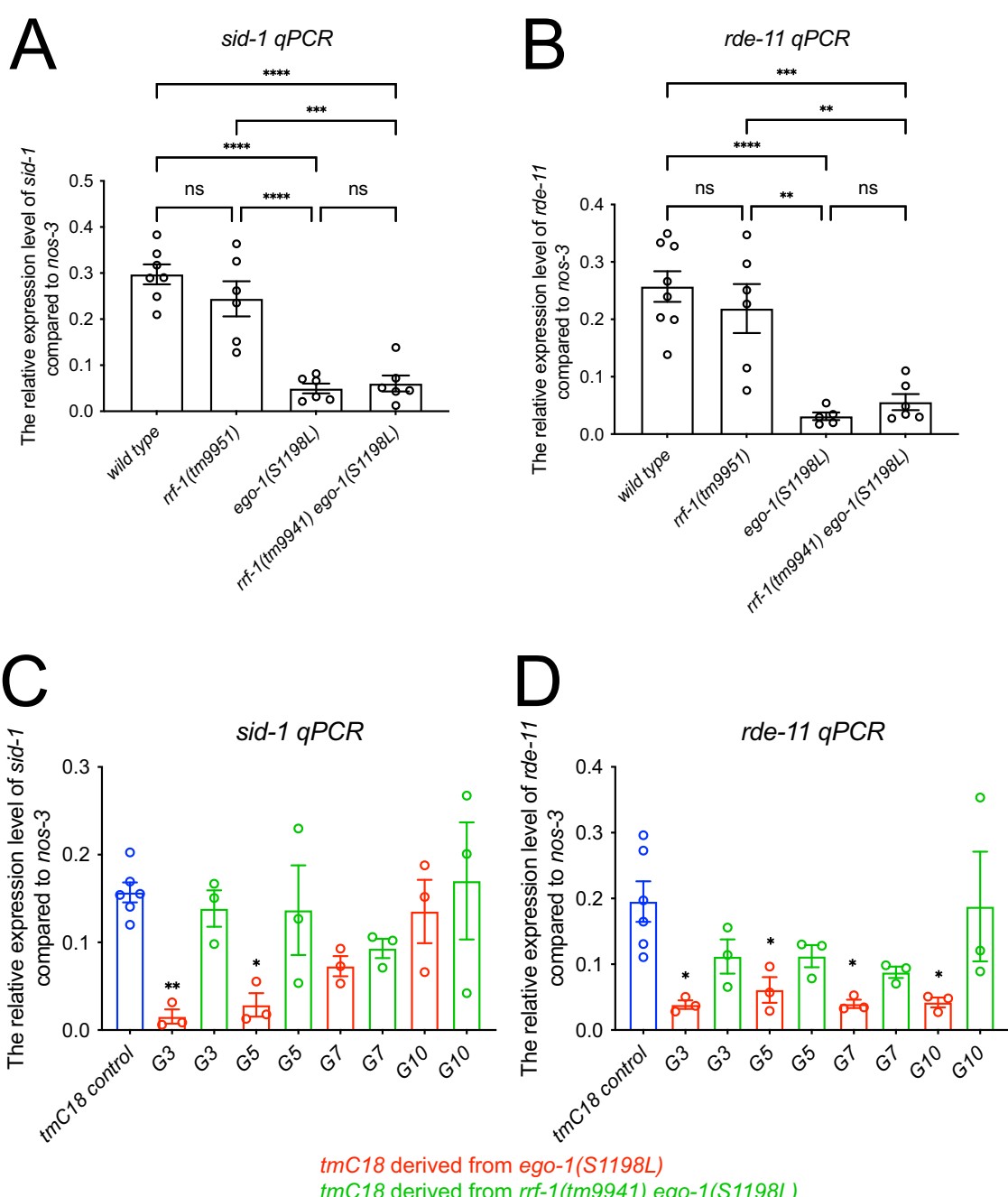

**Figure 6. RRF-1-dependent regulation of *sid-1* and *rde-11* expression across generations in *ego-1(S1198L)* mutants.**

(A, B) RT-qPCR showing the relative expression levels of *sid-1* (A) and *rde-11* (B) mRNA normalized to *nos-3* expression. Error bars represent mean ± SEM. RNA was extracted from young-adult animals grown at 20 °C. (A) Data represent values from seven biological replicates (wildtype) or six biological replicates (other genotypes). (B) Data represent values from eight biological replicates (wildtype), six (*rrf-1(tm9951)* and *rrf-1(tm9941) ego-1(S1198L)*), or five (*ego-1(S1198L)*) biological replicates. *P* values were determined using one-way ANOVA with Šídák's multiple-comparisons test. (A) *** indicates $p < 0.001$ and **** indicates $p < 0.0001$. *P* values for specific comparisons were as follows: wildtype vs. *rrf-1(tm9951)*, $p = 0.5620$; wildtype vs. *ego-1(S1198L)*, $p < 0.0001$; wildtype vs. *rrf-1(tm9941) ego-1(S1198L)*, $p < 0.0001$; *rrf-1(tm9951)* vs. *ego-1(S1198L)*, $p < 0.0001$; *rrf-1(tm9951)* vs. *rrf-1(tm9941) ego-1(S1198L)*, $p = 0.0002$; and *ego-1(S1198L)* vs. *rrf-1(tm9941) ego-1(S1198L)*, $p = 0.9998$. (B) ** indicates $p < 0.01$, *** indicates $p < 0.001$, and **** indicates $p < 0.0001$. *P* values for specific comparisons were as follows: wildtype vs. *rrf-1(tm9951)*, $p = 0.9006$; wildtype vs. *ego-1(S1198L)*, $p < 0.0001$; wildtype vs. *rrf-1(tm9941) ego-1(S1198L)*, $p = 0.0002$; *rrf-1(tm9951)* vs. *ego-1(S1198L)*, $p = 0.0013$; *rrf-1(tm9951)* vs. *rrf-1(tm9941) ego-1(S1198L)*, $p = 0.0034$; and *ego-1(S1198L)* vs. *rrf-1(tm9941) ego-1(S1198L)*, $p = 0.9929$. (C, D) RT-qPCR showing the relative expression levels of *sid-1* (C) and *rde-11* (D) mRNA normalized to *nos-3* expression. Error bars represent mean ± SEM. RNA was extracted from young-adult animals grown at 20 °C. Data represent values from four biological replicates. *P* values were determined using one-way ANOVA with Dunnett's multiple-comparisons test. ns indicates not significant, * indicates $p < 0.05$, ** indicates $p < 0.01$, *** indicates $p < 0.005$, and **** indicates $p < 0.001$. Statistical analyses in (C, D) are in comparison with the *tmC18* control. Source data are available online for this figure.

**Table 2.  Transgenerational effects of *ego-1* MMP mutations on germline exogenous RNAi response at the third and fifth generations.**

| Allele name in the original strain | Amino acid change | Embryonic lethality at G3 (% ± SEM) | | Embryonic lethality at G5 (% ± SEM) | |
|---|---|---|---|---|---|
| | | *pos-1* RNAi | *pop-1* RNAi | *pos-1* RNAi | *pop-1* RNAi |
| wildtype | - | 100 ± 0 | 100 ± 0 | 100 ± 0 | 100 ± 0 |
| gk115401 | E20K | 100 ± 0 | 100 ± 0 | 98.6 ± 1.4 | 100 ± 0 |
| gk720210 | V53I | 99.1 ± 0.45 | 100 ± 0 | 100 ± 0 | 100 ± 0 |
| gk470185 | P161S | 98.29 ± 1.26 | 99.79 ± 0.21 | 98.95 ± 0.53 | 100 ± 0 |
| gk498425 | L246F | 99.17 ± 0.83 | 100 ± 0 | 98.98 ± 0.83 | 100 ± 0 |
| gk115397 | V277M | 99.1 ± 0.9 | 100 ± 0 | 100 ± 0 | 100 ± 0 |
| gk115395 | D304N | 99.04 ± 0.16 | 100 ± 0 | 98.91 ± 0.03 | 100 ± 0 |
| gk837385 | Y313F | 99.75 ± 0.25 | 100 ± 0 | 99.76 ± 0.24 | 100 ± 0 |
| gk115394 | R341K | 99.69 ± 0.31 | 100 ± 0 | 99.71 ± 0.29 | 100 ± 0 |
| gk426642 | R539Q | 41.62 ± 7.92**** | 51.95 ± 24.61**** | 41.21 ± 19.2**** | 81.93 ± 10.26**** |
| gk896494 | L721F | 99.13 ± 0.87 | 100 ± 0 | 100 ± 0 | 100 ± 0 |
| gk115393 | R722H | 100 ± 0 | 100 ± 0 | 100 ± 0 | 100 ± 0 |
| gk481348 | S812F | 100 ± 0 | 99.47 ± 0.53 | 99.55 ± 0.45 | 100 ± 0 |
| gk882383 | C823Y | 97.42 ± 2.58 | 100 ± 0 | 100 ± 0 | 100 ± 0 |
| gk721963 | G844E | 99.4 ± 0.6 | 100 ± 0 | 100 ± 0 | 100 ± 0 |
| gk357146 | V1128E | 100 ± 0 | 100 ± 0 | 100 ± 0 | 100 ± 0 |
| gk925207 | R1165K | 100 ± 0 | 100 ± 0 | 100 ± 0 | 98.55 ± 1.45 |
| gk749674 | G1188E | 100 ± 0 | 100 ± 0 | 99.57 ± 0.43 | 100 ± 0 |
| gk532049 | S1198L | 0.12 ± 0.22**** | 0.22 ± 0.26**** | 0.05 ± 0.07**** | 0.12 ± 0.14**** |
| gk115391 | E1278K | 99.09 ± 0.53 | 100 ± 0 | 100 ± 0 | 100 ± 0 |
| gk317493 | E1397K | 99.06 ± 0.47 | 100 ± 0 | 99.52 ± 0.48 | 100 ± 0 |
| gk115390 | K1469E | 100 ± 0 | 100 ± 0 | 99.54 ± 0.46 | 100 ± 0 |
| gk864727 | N1577S | 100 ± 0 | 100 ± 0 | 100 ± 0 | 100 ± 0 |
| gk540555 | G1610R | 100 ± 0 | 100 ± 0 | 100 ± 0 | 100 ± 0 |

The experimental scheme is basically the same as that shown in Fig. 5A, where "mut" indicates each genotype shown in the table. Data for gk532049 are the same as shown in Fig. 4B (*pos-1* RNAi) and 4C (*pop-1* RNAi). All data represent the values of three technical replicates. *P* values were determined using one-way ANOVA with Dunnett's multiple-comparisons test. **** indicates *p* value < 0.001 compared to control animals derived from wild-type strain.

disruption of RNAi-related molecules depends on the target gene, and a recent study suggests that these selective effects arise from differences in the RNA metabolism of the target genes (Knudsen-Palmer et al, 2024). The differential effects of RNAi regulatory factor mutations on these RNAi efficiency may reflect differences in the metabolic regulation of these mRNAs. One possible mechanism may be differences in their susceptibility to HRDE-1-dependent RNA silencing, with the *pop-1* gene being more preferentially targeted by HRDE-1. A comprehensive understanding of the metabolic regulation of germline mRNAs, including their specific interactions with RNAi pathways, will be important to gain deeper insights into the RNAi mechanism in the future.

It is known that *ego-1* is involved in both germline exo-RNAi and germline development; however, it is not clear whether germline abnormalities and exo-RNAi defects occur independently or concurrently. Here, we found that some *ego-1* mutants were fertile, despite germline exo-RNAi defects. In the *ego-1(S1198L)* mutant, germline exo-RNAi defects were observed without reproductive developmental abnormalities, but abnormal distributions of PGL-1 and ZNFX-1 were observed. This suggests that the

inhibition of EGO-1 does not necessarily cause RNAi defects due to reproductive developmental abnormalities; rather, the abnormal function of P granules and/or Z granules may contribute to RNAi defects, along with reduced RdRP activity of EGO-1.

Mutations with germline exo-RNAi defects have amino acid substitutions at residues that are predicted as "deleterious" mutations by the PROVEAN and PolyPhen-2 prediction programs. C823Y and R539Q are substitutions located within the RdRP domain but in close proximity, whereas V1128E and S1198L are substitutions located outside the RdRP domain. S1198 and R539, whose mutations are associated with strong germline exo-RNAi defects, were predicted to be inside the protein using AlphaFold 3D structure prediction. These mutations are relatively close to each other: the distance between R539 and S1198 is 15.1 Å, between C823 and S1198 is 32.49 Å, and between V1128 and S1198 is 29.64 Å (Fig. EV8). This proximity raises the possibility that these mutations may similarly affect RdRP activity and/or influence interactions with other molecules, such as ZNFX-1, through allosteric effects. It is intriguing that these four mutations correspond to amino acids conserved in RRF-1. Future studies,

**Table 3.** The relative expression levels of *sid-1* and *rde-11* mRNA normalized to *nos-3* expression.

| Allele name | Amino acid change | *sid-1* relative expression (mean ± SD) | *rde-11* relative expression (mean ± SD) |
|---|---|---|---|
| wildtype | - | 0.4 ± 0.06 | 0.35 ± 0.1 |
| gk115401 | E20K | 0.5 ± 0.13 | 0.34 ± 0.04 |
| gk720210 | V53I | 0.37 ± 0.05 | 0.29 ± 0.06 |
| gk470185 | P161S | 0.46 ± 0.09 | 0.33 ± 0.06 |
| gk498425 | L246F | 0.41 ± 0.14 | 0.4 ± 0.13 |
| gk115397 | V277M | 0.41 ± 0.13 | 0.35 ± 0.16 |
| gk115395 | D304N | 0.33 ± 0.03 | 0.35 ± 0.06 |
| gk837385 | Y313F | 0.32 ± 0.05 | 0.26 ± 0.04 |
| gk115394 | R341K | 0.6 ± 0.19 | 0.38 ± 0.07 |
| gk426642 | R539Q | 0.28 ± 0.02 | 0.03 ± 0.01**** |
| gk896494 | L721F | 0.47 ± 0.18 | 0.41 ± 0.11 |
| gk115393 | R722H | 0.56 ± 0.33 | 0.27 ± 0.06 |
| gk481348 | S812F | 0.38 ± 0.09 | 0.25 ± 0.06 |
| gk882383 | C823Y | 0.18 ± 0.04 | 0.21 ± 0.05 |
| gk721963 | G844E | 0.31 ± 0.05 | 0.31 ± 0.07 |
| gk357146 | V1128E | 0.18 ± 0.05 | 0.3 ± 0.03 |
| gk925207 | R1165K | 0.6 ± 0.24 | 0.36 ± 0.1 |
| gk749674 | G1188E | 0.2 ± 0.07 | 0.21 ± 0.07 |
| gk532049 | S1198L | 0.015 ± 0.011**** | 0.039 ± 0.009**** |
| gk115391 | E1278K | 0.56 ± 0.35 | 0.36 ± 0.12 |
| gk317493 | E1397K | 0.47 ± 0.16 | 0.29 ± 0.05 |
| gk115390 | K1469E | 0.57 ± 0.31 | 0.4 ± 0.09 |
| gk864727 | N1577S | 0.33 ± 0.09 | 0.26 ± 0.08 |
| gk540555 | G1610R | 0.43 ± 0.21 | 0.25 ± 0.05 |

All data represent the values of three biological replicates. Data for *gk532049* are the same as shown in Fig. 6C (*sid-1*) and 6D (*rde-11*). *P* values were determined using one-way ANOVA with Dunnett's multiple-comparisons test. **** indicates *p* value <0.001 compared to wild-type samples.

including chimeric experiments involving EGO-1 and RRF-1, will provide more detailed insights into the underlying mechanisms. The robustness of the network regulating RNAi is crucial not only for biological processes, but also as an experimental tool for gene knockdown. Further elucidation of the working mechanisms of RdRPs could shed light on the developmental and antiviral roles of RNAi and broaden its technological applications.

# Methods

### Reagents and tools table

| Reagent/resource | Reference or source | Identifier or catalog number |
|---|---|---|
| Experimental models | | |
| ego-1(gk540555) I | VC40259 was outcrossed with tmC18[tmIs1200] once | FX34788 |
| ego-1(gk864727) I | VC40886 was outcrossed with tmC18[tmIs1200] once | FX34789 |
| ego-1(gk115390) I | VC20206 was outcrossed with tmC18[tmIs1200] four times | FX34340 |
| ego-1(gk317493) I | VC20319 was outcrossed with tmC18[tmIs1200] once | FX34791 |
| ego-1(gk115391) I | VC30058 was outcrossed with tmC18[tmIs1200] once | FX34792 |
| ego-1(gk532049) I | VC40244 was outcrossed with tmC18[tmIs1200] four times | FX34341 |
| ego-1(gk749674) I | VC40660 was outcrossed with tmC18[tmIs1200] six times | FX34822 |
| ego-1(gk925207) I | VC41006 was outcrossed with tmC18[tmIs1200] once | FX34795 |
| ego-1(gk357146) I | VC20618 was outcrossed with tmC18[tmIs1200] four times | FX34342 |
| ego-1(gk721963) I | VC40613 was outcrossed with tmC18[tmIs1200] six times | FX34823 |
| ego-1(gk882383) I | VC40920 was outcrossed with tmC18[tmIs1200] four times | FX34343 |
| ego-1(gk481348) I | VC40140 was outcrossed with tmC18[tmIs1200] once | FX34799 |
| ego-1(gk115393) I | VC20439 was outcrossed with tmC18[tmIs1200] once | FX34222 |
| ego-1(gk896494) I | VC40951 was outcrossed with tmC18[tmIs1200] once | FX34800 |
| ego-1(gk426642) I | VC30158 was outcrossed with tmC18[tmIs1200] four times | FX34380 |
| ego-1(gk115394) I | VC20474 was outcrossed with tmC18[tmIs1200] once | FX34227 |
| ego-1(gk837385) I | VC40832 was outcrossed with tmC18[tmIs1200] once | FX34802 |
| ego-1(gk115395) I | VC40084 was outcrossed with tmC18[tmIs1200] three times | FX34824 |
| ego-1(gk115397) I | VC20545 was outcrossed with tmC18[tmIs1200] three times | FX34809 |
| ego-1(gk498425) I | VC40175 was outcrossed with tmC18[tmIs1200] two times | FX34347 |
| ego-1(gk470185) I | VC40116 was outcrossed with tmC18[tmIs1200] three times | FX34810 |
| ego-1(gk720210) I | VC40611 was outcrossed with tmC18[tmIs1200] once | FX34805 |
| ego-1(gk115401) I | VC40050 was outcrossed with tmC18[tmIs1200] once | FX31735 |
| rrf-1(pk1417) I | from CGC | MAH23 |
| rrf-1(tm9951) I | This study, created by CRISPR-Cas9. F26A3: 6169/6170-7212/7213(1043 bp deletion) | FX9951 |
| rrf-1(tm9941) ego-1(gk532049) I | This study, created by CRISPR-Cas9. F26A3:5756/5757-7220/7221(1464 bp deletion) | FX9941 |
| mut-15(tm1358) V | from NBRP | FX01358 |
| mut-14(pk738) V | from CGC | NL1838 |
| tmC18[tmIs1200] I | from NBRP | FX30167 |
| ego-1(tm521)/tmC18[tmIs1200] I | from NBRP | FX33930 |
| hrde-1(tm1200) III | FX01200 hrde-1(tm1200) was outcrossed with N2 twice, from NBRP | FX34449 |
| ego-1(gk532049) I; hrde-1(tm1200) III | This study | FX34348 |
| hrde-1(tor125[GFP::3xFLAG::hrde-1]) III | from CGC | JMC231 |
| ego-1(gk532049)/tmC18[tmIs1200] I; gfp::3xflag::hrde-1(tor125) III | This study | FX34770 |
| znfx-1(gg544[3xflag::gfp::znfx-1]) II; pgl-1(gg547[pgl-1::3xflag::tagRFP]) IV | from CGC | YY968 |
| ego-1(gk532049)/tmC18[tmIs1236] I; znfx-1(gg544[3xflag::gfp::znfx-1]) II; pgl-1(gg547[pgl-1::3xflag::tagRFP]) IV | This study | FX34771 |
| znfx-1(gg561) II | from CGC | YY996 |

| Reagent/resource | Reference or source | Identifier or catalog number |
|---|---|---|
| *ego-1(gk532049)/tmC18[tmIs1200] I; znfx-1(gg561) II* | This study | FX34772 |
| *C. elegans* wildtype | CGC | N2 |
| *ego-1(ust351[GFP::ego-1]) I* | from CGC | SHG1675 |
| *ego-1(ust351[GFP::ego-1]) I; pgl-1(gg547[pgl-1::3xflag::tagRFP]) IV* | This study | FX34811 |
| *ego-1(tmIs1358[GFP::ego-1(S1198L)] I; pgl-1(gg547[pgl-1::3xflag::tagRFP]) IV* | This study | FX34812 |
| **Recombinant DNA** | | |
| *rrf-1_sg1_pDD162* | This study | Methods and Protocols |
| *rrf-1_sg2_pDD162* | This study | Methods and Protocols |
| pDD162 | Addgene | Plasmid #47549 |
| pCFJ90 | Addgene | Plasmid #19327 |
| pKD638 | This study | Methods and Protocols |
| pKD640 | This study | Methods and Protocols |
| **Oligonucleotides and other sequence-based reagents** | | |
| primers used for the construction of CRISPR plasmids | Eurofins | Methods and Protocols |
| primers used for quantitative PCR | Eurofins | Methods and Protocols |
| **Chemicals, enzymes and other reagents** | | |
| isopropyl β-D-1-thiogalactopyranoside (IPTG) | Sigma Aldrich | I6758 |
| TRIzol | Invitrogen | Cat: 15596026 |
| SuperScript IV Reverse Transcriptase | Invitrogen | Cat: 18090010 |
| Power SYBR™ Green Master Mix | Applied Biosystems | Cat: 4368702 |
| **Software** | | |
| ImageJ | (Schneider et al, 2012) | SCR_003070 |
| Prism GraphPad 10 for MacOS X | GrapPad Software | N/A |

## Methods and protocols

### Nematode culture and strains

Unless otherwise noted, worms were grown under standard conditions at 20 °C. The wild-type strain was Bristol N2. The food source used was *Escherichia coli* strain OP50-1. The *C. elegans* strain information is summarized in the Reagents and tools table.

### CRISPR/Cas9-mediated genome editing

To generate deletion mutations in the *rrf-1* gene, we used the plasmid-based CRISPR/Cas9 method as described previously (Dejima et al, 2023; Dickinson et al, 2013). We injected a genome editing plasmid mixture containing a multi-guide Cas9/sgRNA plasmid (Dejima et al, 2018) targeting *rrf-1* (50 ng/µl each), an injection marker plasmid pCFJ90 (*Pmyo-2::mCherry*) (5 ng/µl), and an NEB 1 kb DNA ladder (145 ng/µl) into the gonads of wild-type or *ego-1(gk532049)* animals. The deletions were identified by PCR screening and confirmed by Sanger sequencing. The primers used in Cas9/sgRNA plasmid construction are:

*rrf-1*_sg1Fwd 5′- TACATCGTCTCCGTCTCGTTTTTAGAGCTAGAAATAGCAAGT -3′

*rrf-1*_sg1Rev 5′- GACGGAGACGATGTAACGCAAGACATCTCGCAATAGG -3′

*rrf-1*_sg2Fwd 5′- AACACAGAGTCCAACCCGTTTTTAGAGCTAGAAATAGCAAGT -3′

*rrf-1*_sg2Rev 5′- GTTGGACTCTGTGTTCTTCAAGACATCTCGCAATAGG -3′.

To insert the *gfp* sequence at the 3′ end of the *ego-1* gene, a repair template was generated by PCR using a plasmid containing homology arms as a template. The homology arms consisted of 704 bp upstream and 547 bp downstream of the PAM site. The plasmid was constructed by amplifying four fragments using the following primer sets and templates, followed by In-Fusion cloning (Clontech Laboratories, Mountain View, CA). Primers and DNA template for this plasmid construction are:

Fragment#1-primers:

F26A3#R52 5′- TCTTCATGTTGCGAATTTCC -3′

plasmid#R387 5′- TTCTCCTTTACTCATTGTTGCGAGGATTCGGGATA -3′

Fragment#1-DNA template: N2 genomic DNA

Fragment#2-primers:

plasmid#F424 5′- AGGTGGAGGTGGAGCTATGGGGGACGAAGGTTATCG -3′

F26A3#F46 5′- CCGACTTCCGACGTCTAACG -3′

Fragment#2-DNA template: N2 genomic DNA

Fragment#3-primers:

kpr697_FP#F9 5′- ATGAGTAAAGGAGAAGAACTTTTCAC -3′

plasmid#R397 5′- GCTCCACCTCCACCTCCTTTGTATAGTTCATCCATGC -3′

Fragment#3-DNA template: a plasmid derived from pPD95.75

Fragment#4-primers:

plasmid#F425 5′- GACGTCGGAAGTCGGGTGGCACTTTTCGGGGAAAT -3′

plasmid#R398 5′- TTCGCAACATGAAGACAGCTCACTCAAAGGCGGTA -3′

Fragment#4-DNA template: a plasmid derived from pDD162

For amplification of the repair template PCR product, the following primer set was used.

F26A3#R52 5′- TCTTCATGTTGCGAATTTCC -3′

F26A3#F46 5′- CCGACTTCCGACGTCTAACG -3′

To target the *ego-1* and *dpy-10* genes, gRNA sequences were inserted into the pDD162 plasmid (Dickinson et al, 2013) using the following primers.

Cas9/sgRNA plasmid for *ego-1* cutting (pKD638):

plasmid#F423 5′- AATCCTCGCAACAATGGGTTTTTAGAGCTAGAAATAGCAAGT -3′

plasmid#R396 5′- ATTGTTGCGAGGATTCGCAAGACATCTCGCAATAGGAG -3′

Cas9/sgRNA plasmid for *dpy-10* cutting (pKD640):

*dpy-10*_sg1#F1 5′- ACCATAGGCACCACGAGGTTTTAGAGCTAGAAATAGCAAGT

*dpy-10*_sg1#R1 5′- CGTGGTGCCTATGGTAGCAAGACATCTCGCAATAGGAG -3′

Repair template for *dpy-10* gene: An ssODN with the following sequence was used as the repair template for the *dpy-10* gene.

ssODN Sequence: 5′- CACTTGAACTTCAATACGGCAAGATGAGAATGACTGGAAACCGTACCGCATGCGGTGCCTATGGTAGCGGAGCTTCACATGGCTTCAGACCAACAGCCTAT -3′

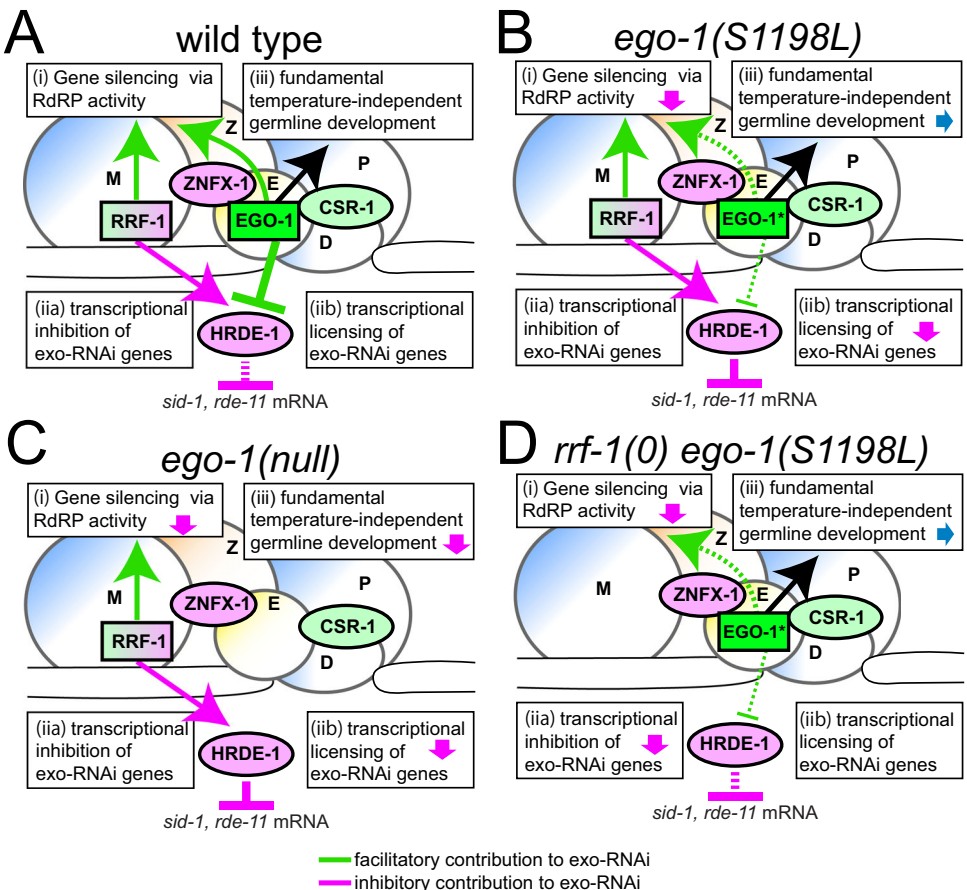

**Figure 7. Model for the regulation of germline exo-RNAi by EGO-1 and RRF-1.**

(A) As noted in previous studies, EGO-1 and RRF-1 share a common role in gene silencing via RdRP activity in germline exo-RNAi (i). However, these RdRPs play distinct roles in the epigenetic regulation of exo-RNAi gene expression (ii). EGO-1 represses the transcriptional inhibition of exo-RNAi genes by HRDE-1, likely through the synthesis of CSR-1-class 22 G RNA (iib). In contrast, RRF-1 is responsible for the synthesis of WAGO-class 22 G RNAs loaded with HRDE-1 and positively acts against the transcriptional inhibition of exo-RNAi genes (iia). Additionally, EGO-1, but not RRF-1, plays an essential role in fundamental temperature-independent germline development, including heterochromatin assembly (iii) (Maine et al, 2005). M represents mutator foci, and Z, E, D, and P represent their respective granules. (B) The ego-1(S1198L) mutation reduces the inhibitory activity against HRDE-1 (green dotted T-line) and RdRP activity directly required for exo-RNAi-mediated gene silencing (arrow with green dotted line). Some functions specific to EGO-1, including its fundamental role in temperature-independent germline development, remain unchanged in this mutant. (C) In the ego-1 null mutants, both the inhibitory activity against HRDE-1 and the RdRP activity directly required for exo-RNAi-mediated gene silencing are lost. (D) In the double-mutant germline of rrf-1(0) and ego-1(S1198L), HRDE-1-dependent repression of germline exo-RNAi genes does not function well. However, RdRP, which is directly required for exo-RNAi-mediated gene silencing, is inactivated, resulting in germline exo-RNAi defects.

Plasmid and PCR product purification: Plasmids were purified using the PureLink HQ Mini Plasmid Kit (Invitrogen, Carlsbad, CA), and PCR products were purified using the Illustra GFX PCR DNA and Gel Band Purification Kit (GE Healthcare, Little Chalfont, UK). The following injection mix was prepared and microinjected into the gonads of the gk532049 mutant:

50 ng/μL pKD638

50 ng/μL ego-1 repair template

10 ng/μL Pmyo-2::venus

50 ng/μL pKD640

1 μM dpy-10 ssODN

Injected worms were placed on NGM plates and allowed to self-fertilize. Individual roller self-progeny were screened by PCR to determine whether GFP had been inserted, using primers located within gfp and outside the ego-1 repair template. PCR-positive strains were further self-fertilized, and only those in which GFP insertion was detected in the F2 and F3 generations were isolated. Homozygous animals carrying this insertion were fertile and used for imaging analysis. The primers used to detect gfp insertion are:

F26A3#R53 5′- CCCACCATGCACACCAATTAGG -3′

kpr256_FP#R1 5′- CCTTCACCCTCTCCACTGAC -3′

### Outcrossing with a balancer

The ego-1 mutants were obtained from the Caenorhabditis elegans Genetics Center, which is supported by the National Institutes of Health National Center for Research Resources. To outcross the strains, males heterozygous for tmC18[tmIs1200] were crossed with each ego-1 mutant strain. GFP-positive tmC18[tmIs1200] was singled over mutant trans-heterozygous F1 hermaphrodites, and their GFP-negative mutant homozygous progeny (F2) was further singled and propagated.

### Quantitative PCR

Total RNA was isolated from 20 to 30 day-1 adults per replicate using the TRIzol reagent (Invitrogen). Superscript IV reverse transcriptase (Invitrogen) was used for reverse transcription according to the manufacturer's instructions. DNase I-treated total RNA (100 ng) was used for the reverse transcription reaction, and 1/3 of the reverse transcription product was used as the template for the PCR reaction. Quantitative PCR was performed in a 7500 Real-time Thermal cycler (Applied Biosystems) using the Power SYBR master mix (Applied Biosystems) with the following parameters: 95 °C for 10 min, and 40 cycles of 95 °C for 15 s and 55 °C for 60 s. Data were normalized to the *nos-3* gene. Gonads were atrophied in the *ego-1* null mutant. Therefore, *ama-1* was used for normalization instead of *nos-3*.

The primers used are the same primers as those used in (Dejima et al, 2023; Dodson and Kennedy, 2019):

*nos-3*
Y53C12B#F17 5′- GGAGGCTATCGGCAGTATCA -3′
Y53C12B#R19 5′- GTGGCCCTGCTTGAGGATTA -3′
*rde-11*
B0564#F22 5′- GATTTCGGACTCCCTATGTGGAC -3′
B0564#R21 5′- GTAGAGATACAGTCCGTCCAGC -3′
*sid-1*
C04F5#F35 5′- CGGCGAATGAATCCATCTAT -3′
C04F5#R20 5′- CGGGAGCTATGAAGACGAAG -3′
*ama-1*
ama-1_Q#F2 5′- AGATGGACCTCACCGACAAC -3′
ama-1_Q#R2 5′- CTGCAGATTACACGGAAGCA -3′

### RNAi by the feeding method

RNAi was performed as described previously (Dejima et al, 2023; Yoshida et al, 2023). For *pos-1* and *pop-1* feeding RNAi, the L4 worms were cultured on RNAi plates for 24 h. Several adult animals were transferred to new NGM plates, allowed to lay eggs for several hours, and then removed. The percentage of dead eggs was calculated after 24 h. For *gld-1* and *mpk-1* RNAi, 20 to 30 eggs were placed on RNAi plates and cultured for 4 days. Then, 2-day-old adult animals were analyzed. If an animal had fewer than five eggs in its uterus, it was scored as showing a sterile or semi-sterile phenotype.

### Image analysis using a compound microscope and quantification

Animals expressing fluorescent proteins were mounted on 5% agar pads in the presence of 10–25 mM $NaN_3$ and imaged with a BX51 microscope equipped with a DP80 CCD camera (Olympus Optical Co., Ltd.). Images were captured at 40× magnification with a resolution of $1360 \times 1024$ pixels, and identical settings and exposure times were applied for each fluorescent protein. Fluorescence intensity quantification was performed using custom image analysis scripts implemented in Fiji. To determine the regions of interest (ROIs) for the particles of GFP::ZNFX-1 and tagRFP::PGL-1, the following processing steps were performed. A duplicate was created for each image, the Fast Fourier Transform (FFT) was applied, and a rectangular region in the center of the FFT-applied image with dimensions of 30×30 pixels was cleared. The resulting image, resembling noise-reduced images perceptible to the human eye, was obtained by applying inverse FFT to transform the image back into the spatial domain. Then, background subtraction (rolling ball algorithm with a rolling radius of 50 pixels), thresholding (Huang dark method), and morphological operations (erosion and watershed) were employed to segment individual puncta. The resulting ROIs were then applied to the original image for intensity measurement. To analyze HRDE-1::GFP, a single nucleus with the optimal focus within the pachytene region was arbitrarily chosen by selecting a square region ($104 \times 104$ pixels) per animal. A rolling ball algorithm with a rolling radius of 20 pixels was applied to subtract background fluorescence. Then, automatic thresholding using the Huang method was performed to convert the image into a binary representation, highlighting the ROI. The image was converted to a binary mask to isolate the nucleus from the background. A selection corresponding to the segmented nucleus was created for subsequent intensity measurements. This selection was then applied to the original image, and the fluorescence intensity in the nucleus was quantified. Cytosolic intensity was calculated by subtracting the nuclear intensity from the overall intensity.

### Image analysis using a confocal laser scanning microscope and quantification

Day 1 adult worms expressing fluorescent proteins were mounted on 5% agar pads in the presence of 1% phenoxyisopropanol and imaged using a Zeiss LSM 900 laser scanning confocal microscope equipped with ZEN software (Zeiss) for image analysis at 63x magnification. Airyscan images were acquired using the Airyscan module on the LSM 900 laser scanning confocal microscope and processed in ZEN after acquisition.

### Statistics

Statistical analyses were performed using Prism 10 (GraphPad Software, Inc.). All bar graphs show the mean ± SEM. Comparisons between more than two groups were performed using one-way ANOVA, while pairwise comparisons were conducted using Student's *t*-test. Dunnett's multiple-comparisons test was used for multiple-comparisons. Data were considered statistically significant at a *p* value of less than 0.05. * indicates $p < 0.05$, ** indicates $p < 0.01$, *** indicates $p < 0.005$, and **** indicates $p < 0.001$.

## Data availability

This study includes no data deposited in external repositories.

The source data of this paper are collected in the following database record: biostudies:S-SCDT-10_1038-S44319-025-00543-0.

## Peer review information

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

## Acknowledgements

We thank the members of the Mitani Laboratory for their support. In this research work we used Institute for Comprehensive Medical Sciences (ICMS), Tokyo Women's Medical University. Some strains were provided by the CGC, which was funded by the NIH Office of Research Infrastructure Programs (P40 OD010440). This work was supported by Grants-in-Aid for Scientific Research to KD (20K06561 and 24K09370) and SM (20H03422), and a grant from the Takeda Science Foundation.

## Author contributions

**Katsufumi Dejima**: Conceptualization; Funding acquisition; Investigation; Methodology; Writing—original draft; Writing—review and editing. **Keita Yoshida**: Investigation; Writing—review and editing. **Shohei Mitani**: Conceptualization; Supervision; Writing—original draft; Project administration; Writing—review and editing.

Source data underlying figure panels in this paper may have individual authorship assigned. Where available, figure panel/source data authorship is listed in the following database record: biostudies:S-SCDT-10_1038-S44319-025-00543-0.

## Disclosure and competing interests statement

The authors declare no competing interests.

# Expanded View Figures

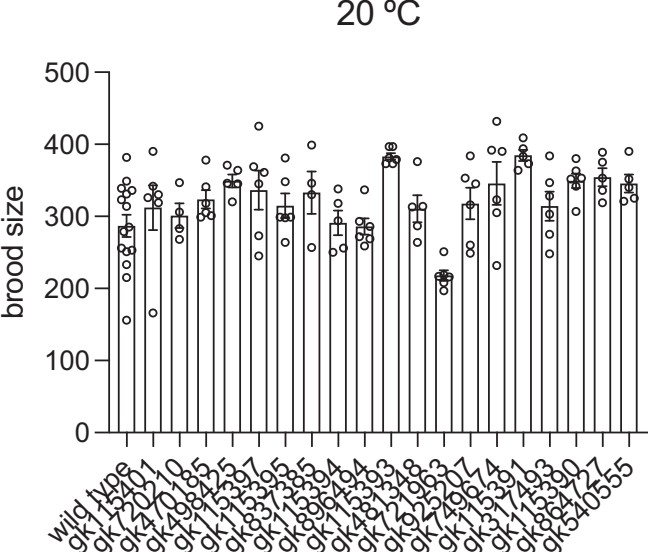

**Figure EV1.** *ego-1* **gk mutants show normal brood size.**

Brood size analysis of the indicated mutant animals that were cultured at 20 °C. The number of biological replicates (*n*) for each genotype was as follows: *gk115401*, *gk470185*, *gk115397*, *gk115395*, *gk896494*, *gk115393*, *gk721963*, *gk925207*, *gk749674*, *gk317493*, and *gk115390* (*n* = 6); *gk720210* and *gk837385* (*n* = 4); *gk498425*, *gk115394*, *gk481348*, *gk115391*, *gk864727*, and *gk540555* (*n* = 5). Error bars represent mean ± SEM. Wild-type data are shown for reference and are the same as those shown in Fig. 1D. Source data are available online for this figure.

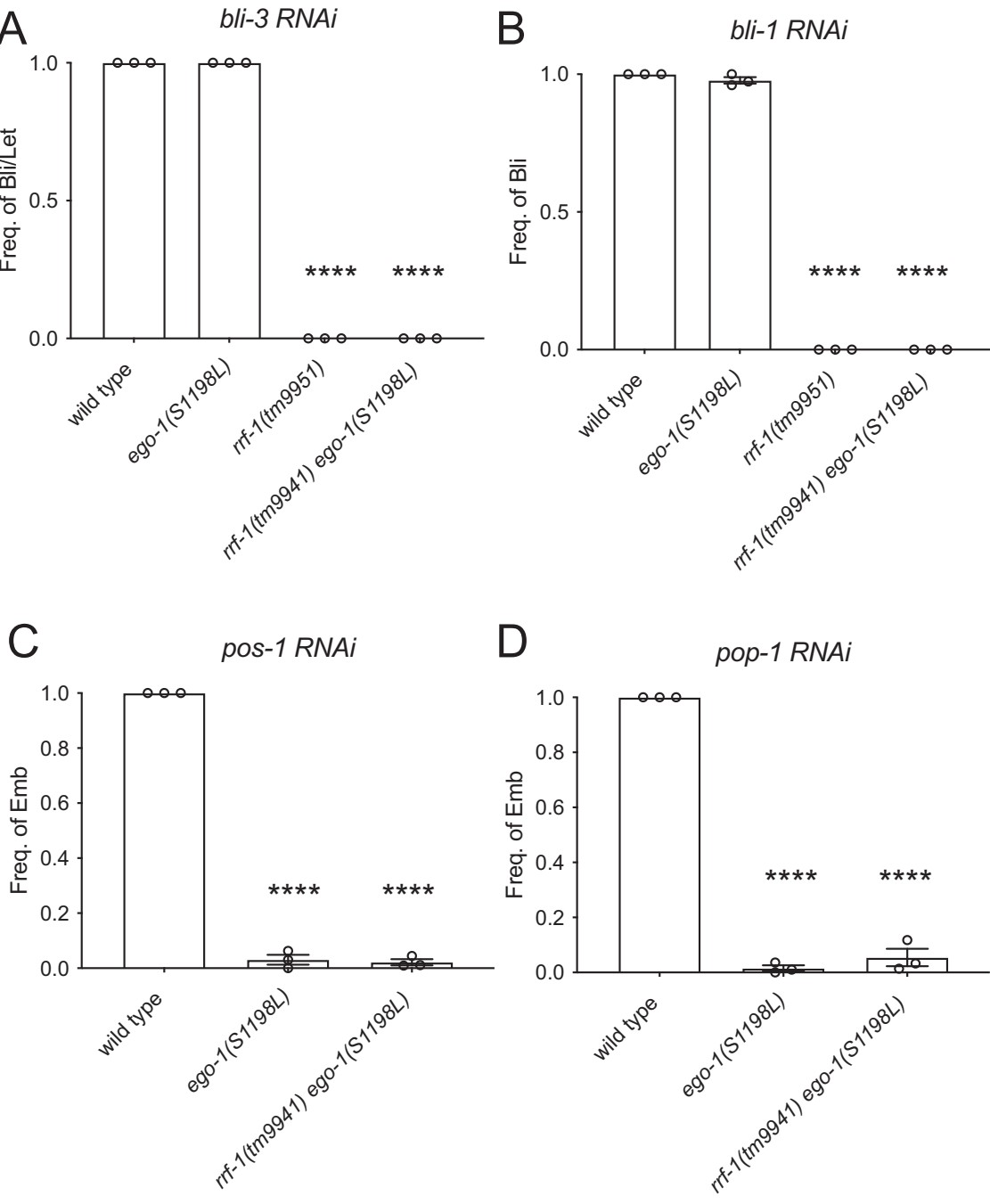

**Figure EV2. *ego-1(S1198L)* is sensitive to somatic RNAi.**

(A, B) Quantification of the frequency of the indicated mutant animals showing the expected phenotype under *bli-3* (A) and *bli-1* (B) feeding RNAi. Error bars represent mean ± SEM. Data represent values from three technical replicates. (C, D) Quantification of the frequency of indicated mutant animals showing the Emb phenotype under *pos-1* (C) and *pop-1* (D) feeding RNAi. Error bars represent mean ± SEM. Data represent values from three technical replicates. *P* values were determined using one-way ANOVA with Dunnett's multiple-comparisons test, compared to wildtype. **** indicates *p* < 0.0001. *P* values for specific comparisons were as follows: (A) *ego-1(S1198L)*, *p* = 0.4044; *rrf-1(tm9951)*, *p* < 0.0001; and *rrf-1(tm9941) ego-1(S1198L)*, *p* < 0.0001. (B) *ego-1(S1198L)*, *p* = 0.0689; *rrf-1(tm9951)*, *p* < 0.0001; and *rrf-1(tm9941) ego-1(S1198L)*, *p* < 0.0001. (C) *ego-1(S1198L)*, *p* < 0.0001; *rrf-1(tm9941) ego-1(S1198L)*, *p* < 0.0001. (D) *ego-1(S1198L)*, *p* < 0.0001; and *rrf-1(tm9941) ego-1(S1198L)*, *p* < 0.0001. Source data are available online for this figure.

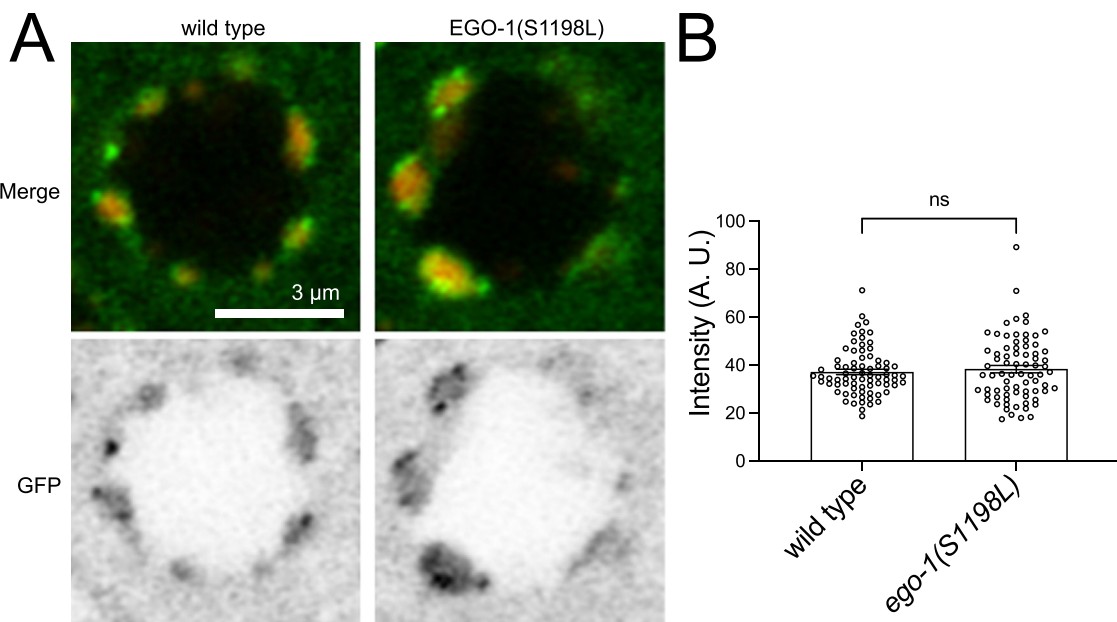

**Figure EV3.   GFP::EGO-1(S1198L) shows similar expression to that of GFP::EGO-1.**

(A) Super-resolution image of a single pachytene nucleus of wild-type EGO-1 and EGO-1(S1198L) tagged with GFP (green) in late pachytene cells of 1-day-old adults. tagRFP::PGL-1 was used as a germ granule marker (red). (B) Bar graph shows the quantification of granular GFP::EGO-1 and GFP::EGO-1(S1198L) fluorescence signal intensity, with $n = 75$ for each genotype (25 animals, three ROIs (Nucleus and surrounded region) each). $P$ values were determined using a Student's $t$-test. ns indicates not significant. Source data are available online for this figure.

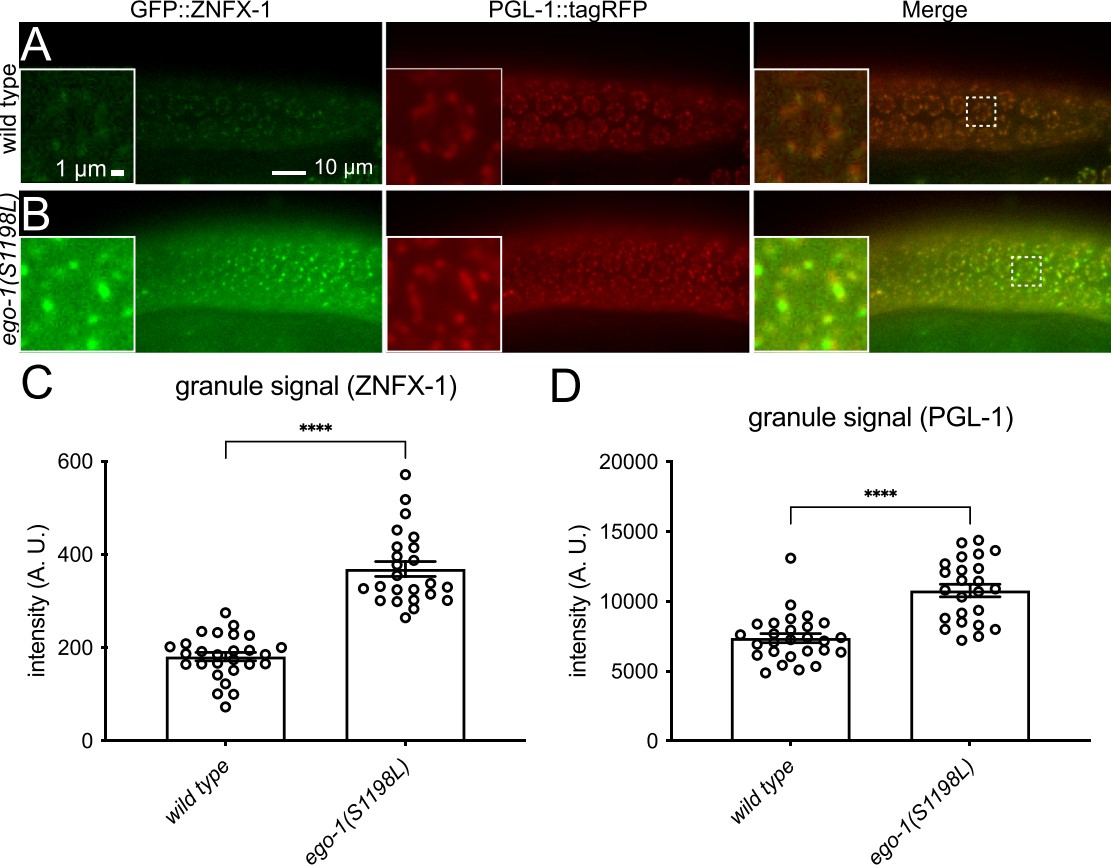

**Figure EV4. Expression of GFP::ZNFX-1 and PGL-1::tagRFP in *ego-1(S1198L)*.**

(A, B) Fluorescence images of germ cells showing the localization of GFP::ZNFX-1 and PGL-1::tagRFP in the wild-type (A) and *ego-1(S1198L)* (B). (C, D) Bar graphs show the quantification of granular GFP::ZNFX-1 (C) and PGL-1::tagRFP (D) fluorescence signal intensity. Error bars represent mean ± SEM. Figure EV4C shows data for wildtype ($n = 27$) and *ego-1(S1198L)* ($n = 24$) animals. Figure EV4D shows data for wildtype ($n = 24$) and *ego-1(S1198L)* ($n = 21$) animals. Statistical significance was determined using two-tailed unpaired *t*-tests. In all cases, the difference was significant (****$p < 0.0001$). Source data are available online for this figure.

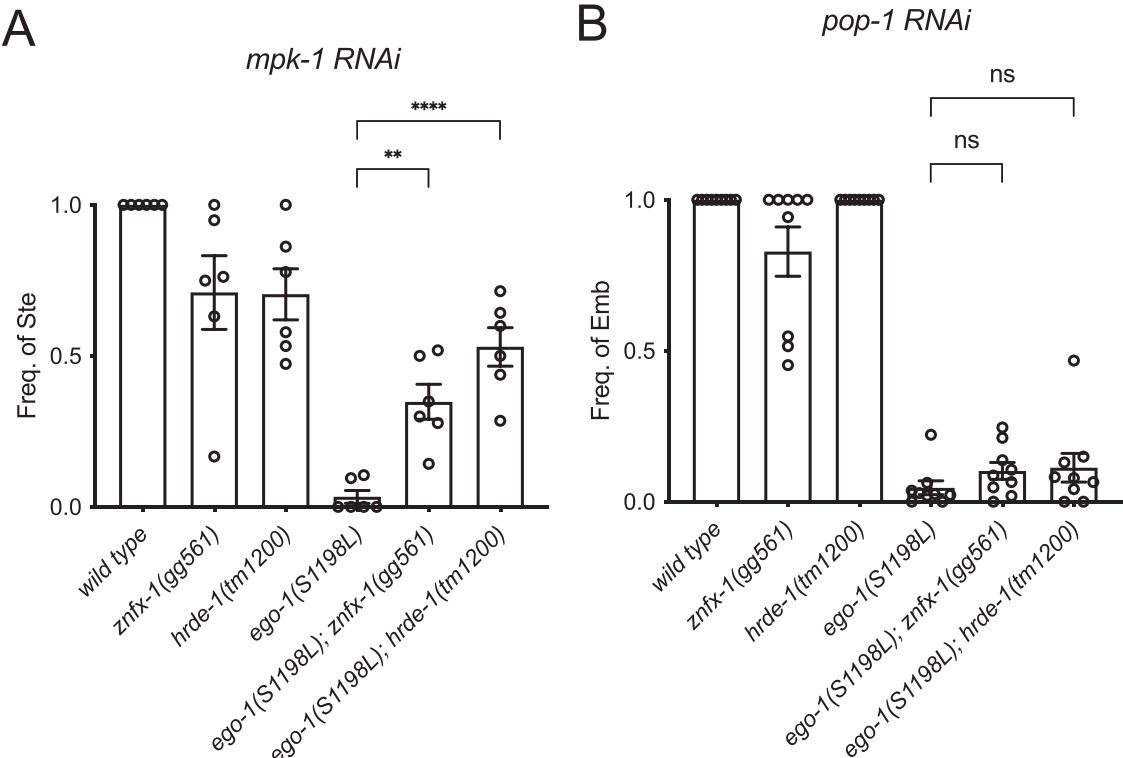

**Figure EV5. Defects in germline exo-RNAi targeting *mpk-1* in *ego-1(S1198L)* are suppressed by *hrde-1* and *znfx-1* mutations.**

(A) Quantification of the frequency of indicated mutant animals showing the Ste phenotype under *mpk-1* feeding RNAi. Error bars represent mean ± SEM. Data represent values from six technical replicates. P values were determined using one-way ANOVA with Dunnett's multiple-comparisons test. **** and ** indicate p value < 0.001 and < 0.01 (0.0074). (B) Quantification of the frequency of indicated mutant animals showing the Emb phenotype under *pop-1* feeding RNAi. Error bars represent mean ± SEM. Data represent values from nine technical replicates. P values were determined using one-way ANOVA with Dunnett's multiple-comparisons test compared to wildtype. **** indicates p value < 0.001. p < 0.0001 for V1128E, R539Q, and S1198L. For C823Y, p = 0.7426. Source data are available online for this figure.

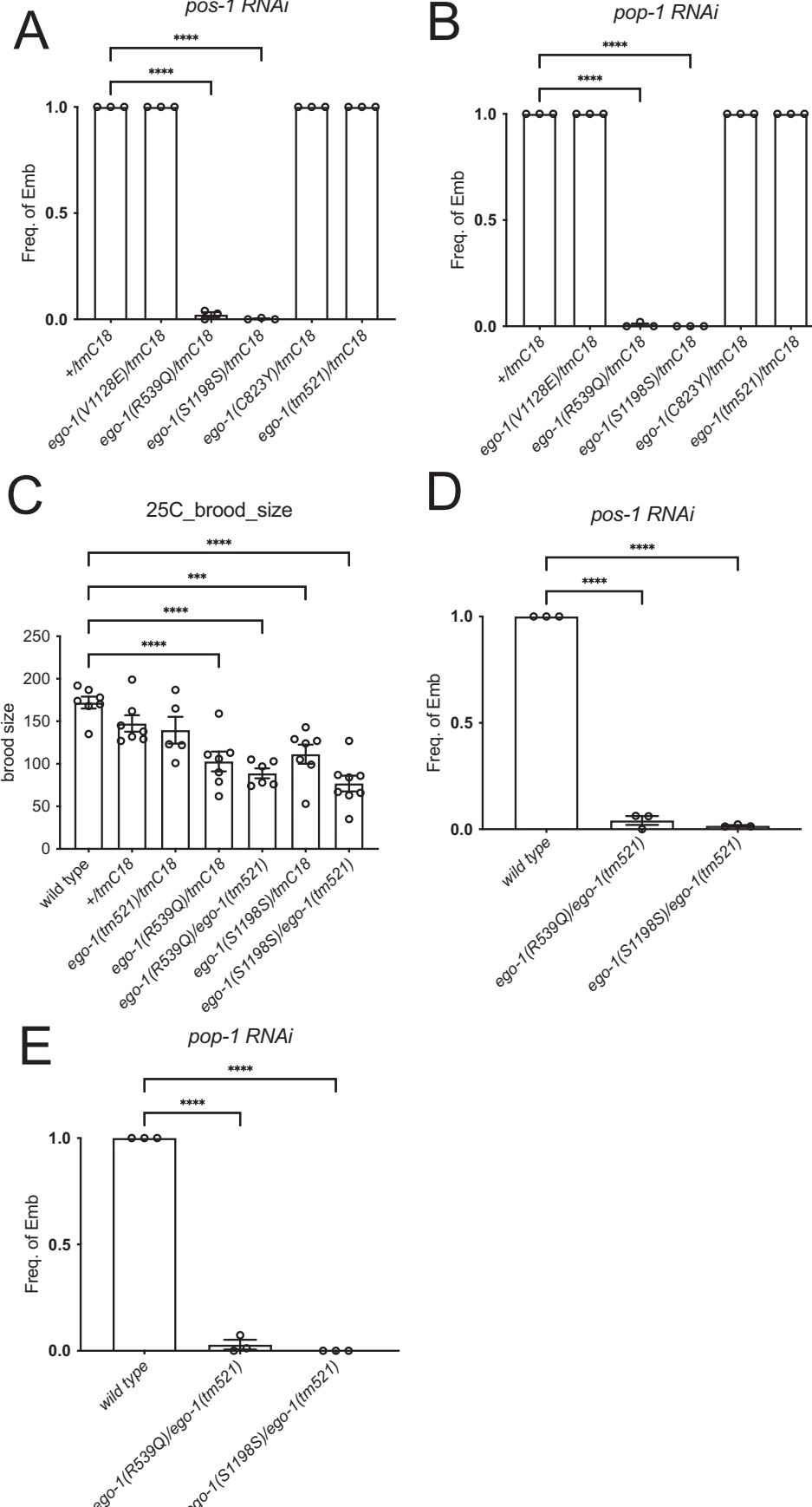

**Figure EV6. RNAi-defective phenotype caused by *ego-1(R539Q)* and *ego-1(S1198L)* heterozygous animals.**

(A) Quantification of the frequency of indicated mutant animals showing the Emb phenotype under *pos-1* feeding RNAi. Error bars represent mean ± SEM. Data represent values from three technical replicates. (B) Quantification of the frequency of indicated mutant animals showing the Emb phenotype under *pop-1* feeding RNAi. Error bars represent mean ± SEM. Data represent values from three technical replicates. In A and B, statistical analysis was performed using one-way ANOVA with Dunnett's multiple-comparisons test, compared to the +/tmC18 control. **** indicates $p < 0.0001$. (A) Significant differences were observed for *R539Q/tmC18* and *S1198L/tmC18* ($p < 0.0001$ for both). No significant differences were found for *V1128E/tmC18*, *C823Y/tmC18*, or *ego-1(tm521)/tmC18* ($p > 0.9999$ for all). (B) Significant differences were observed for *R539Q/tmC18* and *S1198L/tmC18* ($p < 0.0001$ for both). No significant differences were found for *V1128E/tmC18*, *C823Y/tmC18*, or *ego-1(tm521)/tmC18* ($p > 0.9999$ for all). (C) Brood size analysis of the indicated mutant animals that were cultured at 25 °C. Error bars represent mean ± SEM. Data represent values from wildtype ($n = 7$), +/tmC18 ($n = 7$), *ego-1(tm521)/tmC18* ($n = 5$), *R539Q/tmC18* ($n = 7$), *R539Q/ego-1(tm521)* ($n = 6$), and *S1198L/tmC18* ($n = 7$). Statistical analysis was performed using one-way ANOVA with Dunnett's multiple-comparisons test, compared to wild-type control. *** indicates $p < 0.001$ and **** indicates $p < 0.0001$. Significant differences were observed for all *R539Q* and *S1198L* genotypes shown. No significant differences were found for +/tmC18 or *ego-1(tm521)/tmC18*. (D) Quantification of the frequency of indicated mutant animals showing the Emb phenotype under *pos-1* feeding RNAi. Error bars represent mean ± SEM. Data represent values from three technical replicates. (E) Quantification of the frequency of indicated mutant animals showing the Emb phenotype under *pop-1* feeding RNAi. Error bars represent mean ± SEM. Data represent values from three technical replicates. In (D, E), statistical analysis was performed using one-way ANOVA with Dunnett's multiple-comparisons test, compared to wildtype. (D) *** indicates $p < 0.001$ and **** indicates $p < 0.0001$. Significant differences were observed for all *R539Q* and *S1198L* genotypes shown. No significant differences were found for +/tmC18 or *ego-1(tm521)/tmC18*. (E) **** indicates $p < 0.0001$. *P* values for specific comparisons were as follows: *R539Q/ego-1(tm521)*, $p < 0.0001$; and *S1198L/ego-1(tm521)*, $p < 0.0001$. Source data are available online for this figure.

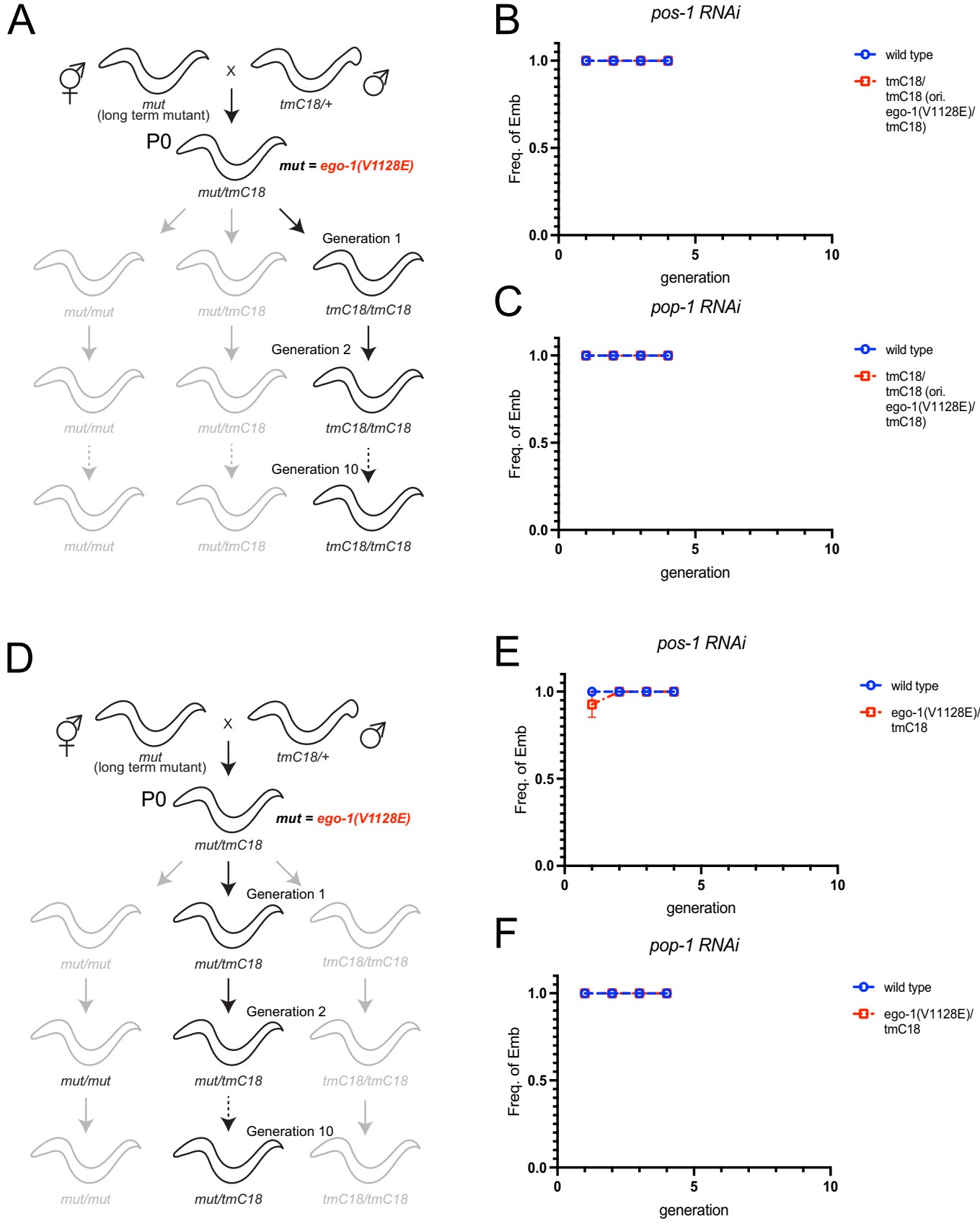

◄ **Figure EV7. Transgenerational effects of *ego-1(V1128E)* on germline exo-RNAi.**

(A, D). Schematics of genetic crosses. P0 is the first generation in which cross-progenies are selected. (B, C) Quantification of the frequency of the indicated mutant animals showing the Emb phenotype under *pos-1* (B) and *pop-1* (C) feeding RNAi in the indicated generation. Data represent values from four biological replicates. These experiments were conducted using a similar experimental setup as in Fig. 5. Source data are available online for this figure.

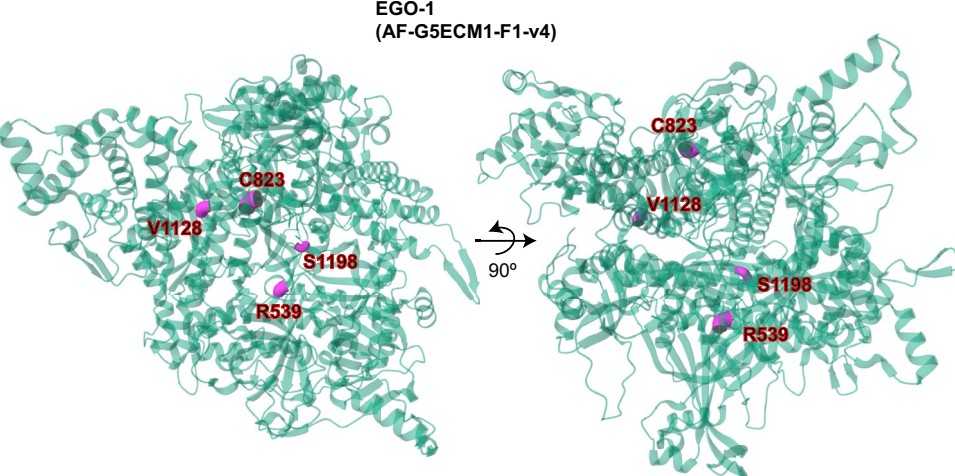

**EGO-1
(AF-G5ECM1-F1-v4)**

**Figure EV8. Structure of wild-type EGO-1 with highlighted mutant sites (C623, V1128, R539, and S1198).**

The 3D structure of the EGO-1 protein was predicted by Alphafold. Image was obtained from Uniprot.

