## [Peer Review File · EMBO Reports]

Dual roles of EGO-1 and RRF-1 in regulating germline exo-RNAi efficiency in *Caenorhabditis elegans*

Katsufumi Dejima, Keita Yoshida, and Shohei Mitani

Corresponding author(s): Shohei Mitani (mitani.shohei@twmu.ac.jp)

Review Timeline:

Transfer Date:	25th Mar 25
Editorial Decision:	2nd May 25
Revision Received:	15th Jun 25
Editorial Decision:	3rd Jul 25
Revision Received:	9th Jul 25
Accepted:	28th Jul 25

Editor: Esther Schnapp

Transaction Report: This manuscript was transferred to EMBO reports following peer review at Review Commons.

Review
COMMONS

Reviewer #1 (Evidence, reproducibility and clarity (Required)):

The study conducted by Katsufumi Dejima and colleagues represents an advance in understanding the multiple roles of RdRPs in *C. elegans* germ cells. EGO-1 is an essential RdRP that is required for multiple aspects of *C. elegans* germline development and efficient RNAi of germline-expressed genes. Yet, currently there is a lack of sufficient genetic mutants to differentiate the multiple biological functions of EGO-1. In this study, the authors examined a large number of non-null alleles for *ego-1* gene and identified four alleles that affect exogenous RNAi, while does not compromise fertility. The authors then focused on the allele *ego-1*(S1198L), examined its influence on germ granule compartments and investigated the molecular mechanism of EGO-1's involvement in feeding RNA interference. Together, their work reveal an extensive interdependent RdRP network that is responsible for regulating exo-RNAi in the germline.

Overall, this is a well-executed study that uncovers the molecular mechanism of EGO-1' function in germline RNAi response and the multiple roles of EGO-1 and RRF-1 in regulating germline RNAi. The findings are poised to have an impact on RNAi research fields.

I have a few comments below. While they are largely minor, addressing them would further enhance the manuscript's clarity and impact.

1. A systemic analysis of the influence of these *ego-1* alleles on fertility can provide valuable information on further studies on EGO-1's functions in fertility.
2. The genotype of JMC231 is *hrde-1*(*tor125*[GFP::3xFLAG::*hrde-1*]) III. In line 245 and 551, HRDE-1::GFP is typed. typo?
3. In Figure 4C, the fluorescence intensity in *ego-1*(S1198L) appears to be more than twice as high as the wild type animals, yet the mean intensity shows only mildly upregulated in Figure 4D. Is the images representative?
4. A brief introduction of tmC18 in the legend of Figure 6 would be friendly to readers.
5. In the discussion section, a detailed summary of three recent published papers about the "phenotypic hangover" phenotype would help to understand how EGO-1 contribute to feeding RNAi. (Dodson & Kennedy, 2019; Lev et al., 2019; Ouyang et al., 2019).
6. Has the authors examined the cellular localization of EGO-1(S1198L) ? Construction

of gfp::ego-1(S1198L) animals would provide this information.

Reviewer #1 (Significance (Required)):

Strength: Enough genetic alleles to differentiate the multiple biological functions of EGO-1.

Limitations: Whether mutant alleles affect siRNA production is unknown.

Advance: The multiple functions of RdRp protein were analyzed through genetic means.

Audience: Basic research, small RNA community and *C. elegans* community

My expertise: small RNA and germ granule.

Reviewer #2 (Evidence, reproducibility and clarity (Required)):

Summary

EGO-1 is a *C. elegans* RNA-directed RNA polymerase well known to amplify small-interfering (si) RNA in the germline and to be required for germline development. The authors screened several partial loss-of-function mutations in *ego-1*, identified in the million mutation project collection, and identified one that does not reduce brood size yet is RNAi defective (Rde). Null and most other *ego-1* mutations are completely sterile and strongly Rde. The newly identified allele, which the authors call S1198L, does not disrupt fertility at moderate culture temperatures yet severely disrupts RNAi, indicating that sterility is separable from the Rde phenotype. S1198L mutants do have reduced fertility at elevated culture temperature; this phenotype is enhanced by a *rrf-1* null mutation, suggesting these two RdRPs are redundantly required for fertility under conditions of temperature stress. Using S1198L, they explore the relationship between EGO-1 and expression or function of other components and regulators of the small RNA machinery as well as components of germ granules (RRF-1, HRDE-1, PGL-1, CDE-1/PUP-1, ZNFX-1). One very interesting characteristic of *ego-1*(S1198L) is that it has a dominant RNAi defect, unlike null alleles; therefore, the EGO-1(S1198L) protein may interfere with EGO-1 wt activity. It seems likely that this allele will be useful for exploring additional aspects of EGO-1 activity beyond those included in this report.

Major comments

Key conclusions are convincing, but data and stats need to be clarified in some cases

(see below).

Line 202-211: They found that *znfx-1(-)* partially restored sensitivity of S1198L mutants to *pos-1* RNAi but did not significantly restore *pop-1* RNAi. Later, section 228-243, they provide evidence that *cde-1* and *hrde-1* mutations partially restore sensitivity to *pos-1*, but not *pop-1*, RNAi. The authors should discuss what might be going on here.

Lines 276-279: Confusing as written. The authors do not show RNAi assays for germline genes with *rrf-1*(null) *ego-1*(S1198L) double mutants. They should show these data.

For the wording, I suggest "RRF-1 compensates for partial loss of EGO-1 activity in S1198L with respect to 25{degree sign}C brood size (Fig. #), but not for germline exo-RNAi (Fig. #). Therefore, the defects..."

Minor comments

Throughout, figure legends shown indicate the statistical test used, and the p value must be indicated (e.g., *** indicates p-value of #).

The authors should use consistent nomenclature for the *ego-1* null allele. In Fig. 5 it's listed as " Δ " and elsewhere as tm521.

Line 90: Please include references for the *ego-1* null germline phenotype.

Line 107-109: Wording is confusing. I suggest "Disruption of the E granule, of which EGO-1 is a component, has recently been shown to upregulate sRNA targeting ..."

Line 118-120: Wording is unclear. I suggest "In addition we found that *sid-1* and *rde-11* transcripts in *ego-1*(S1198L) were downregulated, and this effect was suppressed in *hrde-1*, *cde-1*, and *znfx-1* mutants."

Line 121-123: The meaning is unclear. Please clarify what "detached" means in this context.

Line 171-172: Substitute "in the genome" for "in terms of its genomic locus"

Line 207: Substitute "the *pos-1* RNAi defect" for "the Rde phenotype of *pos-1* RNAi"

Line 269: Text says Fig 5A,B, shows restoration to "wt levels," but stats only show significant change from *ego-1*(S1198L). Stats showing comparison with wt should be shown, as well.

The text refers to the wrong figure/panel in some places.

Line 310 references Fig. 6A-C as showing the phenotype of ego-1(+/-) heterozygotes and ego-1(+/+) homozygotes, but only the latter is shown in 6A-C. Heterozygotes are shown in Fig. 6D-F.

Line 350 should reference Fig. 7C, D (not Fig 3A).

Line 380-381: Wording is awkward. I suggest "Additionally, this allele showed synthetic ts sterility with an rrf-1 deletion mutation."

Figure 8: There is a typo in panel C: the allele shown is ego-1(null) not ego-1(S1198).

Reviewer #2 (Significance (Required)):

The paper addresses the mechanisms and activity of small RNA-mediated pathways, including in regulating gene expression and development. The work will be general interest to the large community studying small RNA-mediated gene expression and/or germline development in *C. elegans* and more broadly. The work is significant because it reveals distinct requirements for EGO-1 RdRP in exo-RNAi, germline development under conditions of temperature stress, and germline development more broadly.

I am a *C. elegans* biologist with many decades of experience studying germline development and RNAi-related phenomena.

Reviewer #3 (Evidence, reproducibility and clarity (Required)):

Summary:

Mitani and colleagues' manuscript investigates the role of RNA-dependent RNA polymerase (RdRP) EGO-1 in regulating exogenous RNAi (induced by dsRNA delivery) efficiency in the germline of *C. elegans*. Since the null ego-1 mutation leads to sterility, the authors take advantage of several missense ego-1 mutant strains that are fertile but RNAi-resistant.

Major comments:

- The authors recognize at least two distinct mechanisms of EGO-1 function in regulating exo-RNAi. The first is direct, since EGO-1 RdRP is required for the

production of secondary small RNAs mediating exo-RNAi silencing (this mechanism has been studied for many years), and the second one is indirect, through the role of EGO-1 RdRP in the production of endogenous "licensing" small RNAs that allow germline gene expression, including expression of genes required for exo-RNAi response. In addition, the authors find that the chosen missense mutant strains show a dominant exo-RNAi resistance phenotype, unlike the recessive *ego-1* null.

Although the authors recognize the complex nature of *ego-1* phenotypes and provide a helpful model in Figure 8, I find that not all conclusions are consistent with the presented data. A more rigorous data interpretation and presentation logic is required for publication. Also, some additional simple experiments can be done to enhance the rigor of conclusions.

1. The authors link the direct gene-silencing function of EGO-1 with temperature-sensitive sterility (Figure 8). However, the data in Figure 1 show that the RNAi resistance phenotype and *ts*-sterility are anti-correlated, the most RNAi-resistant *ego-1* alleles are least *ts*-sensitive and vice versa. Therefore, motivating further experiments through the connection between exo-RNAi resistance and *ts*-sterility is not justified, e.g. "the temperature sensitive sterile phenotype is a hallmark of the mutator complex.... which is necessary for exo-RNAi-driven silencing". Also, the claim of the redundancy between *ego-1* and *rrf-1* in controlling *ts*-sterility is not justified. The *ego-1*(V1128E) and (C823Y) alleles show strong *ts*-sterility (Figure 1E), which is not compensated by RRF-1. Therefore, the specific nature of *ego-1*(S1198L) and (R539Q) mutations leads to a higher dependence of endogenous RNAi silencing processes on RRF-1. Remarkably, although the exo-RNAi resistance of these alleles is dominant (Figure EV2 A,B) and clearly distinct from *ego-1* null heterozygous animals, the *ts*-sterility of *ego-1* null heterozygotes and S1198L or R539Q heterozygotes is identical (Figure EV C).

2. The experiments in Figures 6 and Figure 7C,D are the most important findings of this study, showing that EGO-1 has a role in the licensing of genes important for exo-RNAi in the germline (such as *sid-1* and *rde-11*). The apparent persistence of RRF-1-dependent (and presumably HDRE-1-dependent) silencing of *sid-1* and *rde-11* in a genetically wild-type background that correlates with exo-RNAi resistance is remarkable, although not novel (it was shown for mutants defective in P-granules). The use of *ego-1* missense viable background was instrumental in these experiments. However, it is not clear whether the specific nature of *ego-1*(S1198L) mutation also played a role, such as enhanced production of RRF-1-dependent endogenous silencing small RNAs. The *ego-1*(V1128E) allele is an apparent hypomorph, which is viable and exo-RNAi-resistant (Figure 1, EV2A). Performing an experiment shown in Figure 6 with this allele for five generations would be highly illuminating, and either outcome would be

interesting.

3. Conclusions from the experiments in Figures 3 and 4 are not convincing. The imaging data can be moved to supplemental materials. The suppression experiments shown in Figure 4A,B are weak. The effects of *cde-1* mutation are hard to interpret, and these data can be omitted. The *znfx-1* and *hrde-1* loss does not affect resistance to *pop-1*. If the authors want to insist on their model, they should use several additional *exo*-RNAi target genes producing *Emb* (or other) phenotypes and repeat the experiments.

4. The *exo*-RNAi resistance and reduced *sid-1* and *rde-11* expression correlate. The reduction of these *exo*-RNAi factors is a plausible explanation for the epigenetic RNAi resistance shown in Figure 6. However, *ego-1*(S1198L); *hrde-1*(-) P0 is resistant to *pop-1*(RNAi) to a large extent (Figure 4B), while *sid-1* and *rde-11* expression is restored in this double compared to single *ego-1*(S1198L) (Figure 5B). Therefore, *ego-1*(S1198L) *exo*-RNAi resistance is not likely driven to any extent by the misregulation of other RNAi genes. The nature of the (S1198L) mutation is likely to play a major role. Also, surprisingly, *rrf-1*(-) addition to *ego-1*(S1198L) does not restore *sid-1* and *rde-11* expression. Why? The authors do not comment on this.

5. The discussion points about the nature of new EGO-1 missense mutations involving Alpha Fold predictions can be illustrated through Alpha Fold model figures.

6. The authors should consider a model where *ego-1*(S1198L) affects RRF-1 activity such that it is more active in the endogenous RNAi silencing processes at the expense of *exo*-RNAi. This could explain the reduced *ts*-sterility in *ego-1*(S1198L), which is RRF-1-dependent, similar to the better-investigated epigenetic inheritance of *exo*-RNAi resistance. However, the exact mechanism of *ego-1*(S1198L) cannot be explained by genetic methods and is beyond the scope of this study.

- Data and the methods are presented in such a way that they can be reproduced.
- Statistical analyses are adequate.

Minor comments:

- Figure 8C typo: *ego*-(0) is meant to be shown.
- Pak and Fire, *Science*, 2007 should be cited in connection to secondary siRNA production. Ruby and Bartel, *Cell*, 2006 should be cited as the first study that identified 21U-RNAs.

Reviewer #3 (Significance (Required)):

- General assessment:

The strength of this study is in generating reagents suitable for performing experiments that were not feasible with the sterile null mutant. The major finding of the paper is the epigenetic inheritance of resistance to exo-RNAi by the wild-type descendants of ego-1 mutants, which is dependent on rrf-1.

There are numerous weaknesses in the interpretation of other data, which are described in section 1. The study's limitation is the exclusive use of genetic approaches. The effect of the antimorphic point mutations on EGO-1 stability, localization, and interaction with other proteins could have provided more insight into the protein's function.

- The most notable results presented in the paper are very similar to the findings of several groups published in 2019 (Lev et al., Ouyang et al, and Dodson and Kennedy) and, therefore, are not novel. The experimental setup is identical to Dodson and Kennedy; it just uses different mutants. The novel aspect is the opposite relationship between ego-1 and rrf-1, which has not been described before.

- This research will be of interest to *C. elegans* researchers and those following epigenetic phenomena.

- My expertise is in RNAi in *C. elegans* and epigenetics. I have sufficient expertise to evaluate all aspects of the paper.

Full Revision

Manuscript number: RC-2024-02709

Corresponding author(s): Shohei Mitani

[Please use this template only if the submitted manuscript should be considered by the affiliate journal as a full revision in response to the points raised by the reviewers.]

*If you wish to submit a preliminary revision with a revision plan, please use our "Revision Plan" template. **It is important to use the appropriate template to clearly inform the editors of your intentions.**]*

1. General Statements [optional]

This section is optional. Insert here any general statements you wish to make about the goal of the study or about the reviews.

Reviewer #1 (Evidence, reproducibility and clarity (Required)):

*The study conducted by Katsufumi Dejima and colleagues represents an advance in understanding the multiple roles of RdRPs in *C. elegans* germ cells. EGO-1 is an essential RdRP that is required for multiple aspects of *C. elegans* germline development and efficient RNAi of germline-expressed genes. Yet, currently there is a lack of sufficient genetic mutants to differentiate the multiple biological functions of EGO-1. In this study, the authors examined a large number of non-null alleles for *ego-1* gene and identified four alleles that affect exogenous RNAi, while does not compromise fertility. The authors then focused on the allele *ego-1*(S1198L), examined its influence on germ granule compartments and investigated the molecular mechanism of EGO-1's involvement in feeding RNA interference. Together, their work reveal an extensive interdependent RdRP network that is responsible for regulating exo-RNAi in the germline.*

Overall, this is a well-executed study that uncovers the molecular mechanism of EGO-1' function in germline RNAi response and the multiple roles of EGO-1 and RRF-1 in regulating germline RNAi. The findings are poised to have an impact on RNAi research fields.

I have a few comments below. While they are largely minor, addressing them would further enhance the manuscript's clarity and impact.

*1. A systemic analysis of the influence of these *ego-1* alleles on fertility can provide valuable information on further studies on EGO-1's functions in fertility.*

We thank the reviewer for this insightful comment. We scored the brood size of all strains carrying a missense mutation at the *ego-1* locus and added an extended figure showing their brood sizes as Fig. EV1A. Although the strain carrying *gk721963*, which was outcrossed six times with *tmC18*, showed a slightly reduced brood size, other strains showed no significant change in brood size compared to wild-type animals. The original strain carrying *gk721963* has 24 homozygous mutations on chromosome I, where *ego-1* is located. Of these, 15 mutations are in the region covered by *tmC18*, and 9 alleles are not covered. These background mutations may not be unremoved and affect fertility in concert with the *ego-1* mutation. However, we believe that identifying the cause of this slight phenotype is very difficult and not essential to the overall analysis, so we have only presented the scored data for future studies on EGO-1's functions.

2. The genotype of JMC231 is hrde-1(tor125[GFP::3xFLAG::hrde-1]) III. In line 245 and 551, HRDE-1::GFP is typed. typo?

Thank you for pointing this out. We have corrected these for consistency.

*3. In Figure 4C, the fluorescence intensity in *ego-1*(S1198L) appears to be more than twice as high as the wild type animals, yet the mean intensity shows only mildly upregulated in Figure 4D. Is the images representative?*

Thank you for your comment. We agree that the fluorescence intensity in the original wild-type image may not have been representative. To address this concern, we have replaced the wild-type image in Fig. 3C (4C in the previous version) with an image that is more reflective of the average fluorescence intensity observed across the biological replicates.

*4. A brief introduction of *tmC18* in the legend of Figure 6 would be friendly to readers.*

Thank you for your suggestion. We have added statements explaining *tmC18* to the legend of Fig. 5 (Fig. 6 in the previous version) for clarity and to make the experiments more understandable.

5. In the discussion section, a detailed summary of three recent published papers about the "phenotypic hangover" phenotype would help to understand how EGO-1 contribute to feeding RNAi. (Dodson & Kennedy, 2019; Lev et al., 2019; Ouyang et al., 2019).

Thank you for the suggestion. We have incorporated a detailed summary of the "phenotypic hangover" phenotype in the discussion section.

*6. Has the authors examined the cellular localization of EGO-1(S1198L) ? Construction of *gfp::ego-1*(S1198L) animals would provide this information.*

We thank the reviewer for this insightful comment. We have generated the GFP::EGO-1(S1198L) strain and analyzed its subcellular localization and dynamics. These analysis revealed no abnormality in the expression, localization and dynamics of GFP::EGO-1(S1198L) compared to the wild type. The data are shown in Fig. EV3, and a section of the description about this is added to the third section of the Results.

Reviewer #1 (Significance (Required)):

Strength: Enough genetic alleles to differentiate the multiple biological functions of EGO-1.

Limitations: Whether mutant alleles affect siRNA production is unknown.

Advance: The multiple functions of RdRp protein were analyzed through genetic means.

Audience: Basic research, small RNA community and C. elegans community

My expertise: small RNA and germ granule.

Reviewer #2 (Evidence, reproducibility and clarity (Required)):

Summary

EGO-1 is a C. elegans RNA-directed RNA polymerase well known to amplify small-interfering (si) RNA in the germline and to be required for germline development. The authors screened several partial loss-of-function mutations in ego-1, identified in the million mutation project collection, and identified one that does not reduce brood size yet is RNAi defective (Rde). Null and most other ego-1 mutations are completely sterile and strongly Rde. The newly identified allele, which the authors call S1198L, does not disrupt fertility at moderate culture temperatures yet severely disrupts RNAi, indicating that sterility is separable from the Rde phenotype. S1198L mutants do have reduced fertility at elevated culture temperature; this phenotype is enhanced by a rrf-1 null mutation, suggesting these two RdRPs are redundantly required for fertility under conditions of temperature stress. Using S1198L, they explore the relationship between EGO-1 and expression or function of other components and regulators of the small RNA machinery as well as components of germ granules (RRF-1, HRDE-1, PGL-1, CDE-1/PUP-1, ZNFX-1). One very interesting characteristic of ego-1(S1198L) is that it has a dominant RNAi defect, unlike null alleles; therefore, the EGO-1(S1198L) protein may interfere with EGO-1 wt activity. It seems likely that this allele will be useful for exploring additional aspects of EGO-1 activity beyond those included in this report.

Major comments

Key conclusions are convincing, but data and stats need to be clarified in some cases (see below).

*Line 202-211: The found that *znfx-1(-)* partially restored sensitivity of S1198L mutants to *pos-1* RNAi but did not significantly restore *pop-1* RNAi. Later, section 228-243, they provide evidence that *cde-1* and *hrde-1* mutations partially restore sensitivity to *pos-1*, but not *pop-1*, RNAi. The authors should discuss what might be going on here.*

Thank you for your comment. We have added a discussion on the differential restoration of sensitivity to *pos-1* and *pop-1* RNAi in the presence of *znfx-1*, *cde-1*, and *hrde-1* mutations, proposing that this variation may result from differences in the RNA metabolism of these target genes (Knudsen-Palmer et al., 2024). Additionally, we incorporated the results from the additional RNAi experiments targeting *gld-1* and *mpk-1* (as outlined in our response to Reviewer 3, Comment 3), which further support our proposed model. We hope this revision presents a more thorough analysis of the interplay between these mutations and RNAi sensitivity.

*Lines 276-279: Confusing as written. The authors do not show RNAi assays for germline genes with *rrf-1(null)* *ego-1(S1198L)* double mutants. They should show these data.*

Thank you for the feedback. We have added the RNAi assay data for germline genes with *rrf-1(null)* *ego-1(S1198L)* double mutants in Figure EV3C and D.

For the wording, I suggest "RRF-1 compensates for partial loss of EGO-1 activity in S1198L with respect to 25{degree sign}C brood size (Fig. #), but not for germline exo-RNAi (Fig. #). Therefore, the defects..."

Thank you for the suggestion. We have revised the wording as recommended.

Minor comments

*Throughout, figure legends shown indicate the statistical test used, and the p value must be indicated (e.g., *** indicates p-value of #).*

*The authors should use consistent nomenclature for the *ego-1* null allele. In Fig. 5 it's listed as "□" and elsewhere as *tm521*.*

Thank you for pointing this out. We corrected this in the revised manuscript.

*Line 90: Please include references for the *ego-1* null germline phenotype.*

Thank you for your suggestion. We included two references demonstrating the *ego-1* null germline phenotype in the revised manuscript.

Line 107-109: Wording is confusing. I suggest "Disruption of the E granule, of which EGO-1 is a component, has recently been shown to upregulate sRNA targeting ..."

Thank you for the suggestion. We have revised the wording as suggested.

Line 118-120: Wording is unclear. I suggest "In addition we found that sid-1 and rde-11 transcripts in ego-1(S1198L) were downregulated, and this effect was suppressed in hrde-1, cde-1, and znfx-1 mutants."

Thank you for the suggestion. We have revised the wording as suggested.

Line 121-123: The meaning is unclear. Please clarify what "detached" means in this context.

Thank you for the comment. We have revised the sentence to remove the term "detached" for clarity and have instead explicitly described the phenomenon, stating that the RNAi-defective (Rde) phenotype persists over generations in an RRF-1-dependent manner, even in the absence of the original ego-1(S1198L) mutation.

Line 171-172: Substitute "in the genome" for "in terms of its genomic locus"

Thank you for the suggestion. We have revised the wording as suggested.

Line 207: Substitute "the pos-1 RNAi defect" for "the Rde phenotype of pos-1 RNAi"

Thank you for pointing this out. We have revised the text as suggested.

Line 269: Text says Fig 5A,B, shows restoration to "wt levels," but stats only show significant change from ego-1(S1198L). Stats showing comparison with wt should be shown, as well.

Thank you for the comment. We have revised the text to clarify the expression levels and removed the statement about "restoration to wild-type levels" where statistical comparisons were not provided.

The text refers to the wrong figure/panel in some places.

Line 310 references Fig. 6A-C as showing the phenotype of ego-1(+/-) heterozygotes and ego-1(+/+) homozygotes, but only the latter is shown in 6A-C. Heterozygotes are shown in Fig. 6D-F.

Thank you for pointing this out. We have revised the statement accordingly.

Line 350 should reference Fig. 7C, D (not Fig 3A).

Thank you for your suggestion. We have corrected it to Fig. 6C, D (Fig. 7C, D in the previous version) as suggested.

Line 380-381: Wording is awkward. I suggest "Additionally, this allele showed synthetic ts sterility with an rrf-1 deletion mutation."

Thank you for pointing this out. We have revised the text as suggested.

Figure 8: There is a typo in panel C: the allele shown is ego-1(null) not ego-1(S1198).

Thank you for pointing this out. We have updated the allele to ego-1(null) in panel C.

Reviewer #2 (Significance (Required)):

The paper addresses the mechanisms and activity of small RNA-mediated pathways, including in regulating gene expression and development. The work will be general interest to the large community studying small RNA-mediate gene expression and/or germline development in C. elegans and more broadly. The work is significant because it reveals distinct requirements for EGO-1 RdRP in exo-RNAi, germline development under conditions of temperature stress, and germline development more broadly.

I am a C. elegans biologist with many decades of experience studying germline development and RNAi-related phenomena.

Reviewer #3 (Evidence, reproducibility and clarity (Required)):

Summary:

Mitani and colleagues' manuscript investigates the role of RNA-dependent RNA polymerase (RdRP) EGO-1 in regulating exogenous RNAi (induced by dsRNA delivery) efficiency in the germline of C. elegans. Since the null ego-1 mutation leads to sterility, the authors take advantage of several missense ego-1 mutant strains that are fertile but RNAi-resistant.

Major comments:

- The authors recognize at least two distinct mechanisms of EGO-1 function in regulating exo-RNAi. The first is direct, since EGO-1 RdRP is required for the production of secondary small RNAs mediating exo-RNAi silencing (this mechanism has been studied for many years), and the second one is indirect, through the role of EGO-1 RdRP in the production of endogenous "licensing" small RNAs that allow germline gene expression, including expression of genes required for exo-RNAi response. In addition, the authors find that the chosen missense mutant strains show a dominant exo-RNAi resistance phenotype, unlike the recessive ego-1 null.

Although the authors recognize the complex nature of ego-1 phenotypes and provide a helpful model in Figure 8, I find that not all conclusions are consistent with the presented data. A more rigorous data interpretation and presentation logic is required for publication. Also, some additional simple experiments can be done to enhance the rigor of conclusions.

1. The authors link the direct gene-silencing function of EGO-1 with temperature-sensitive sterility (Figure 8). However, the data in Figure 1 show that the RNAi resistance phenotype and *ts*-sterility are anti-correlated, the most RNAi-resistant *ego-1* alleles are least *ts*-sensitive and vice versa. Therefore, motivating further experiments through the connection between *exo*-RNAi resistance and *ts*-sterility is not justified, e.g. "the temperature sensitive sterile phenotype is a hallmark of the mutator complex... which is necessary for *exo*-RNAi-driven silencing". Also, the claim of the redundancy between *ego-1* and *rrf-1* in controlling *ts*-sterility is not justified. The *ego-1*(V1128E) and (C823Y) alleles show strong *ts*-sterility (Figure 1E), which is not compensated by RRF-1. Therefore, the specific nature of *ego-1*(S1198L) and (R539Q) mutations leads to a higher dependence of endogenous RNAi silencing processes on RRF-1. Remarkably, although the *exo*-RNAi resistance of these alleles is dominant (Figure EV2 A,B) and clearly distinct from *ego-1* null heterozygous animals, the *ts*-sterility of *ego-1* null heterozygotes and S1198L or R539Q heterozygotes is identical (Figure EV C).

We thank the reviewer for the insightful comments. We have revised the second section of the Results to simplify the argument by removing descriptions related to WAGO 22G RNA and fertility. This revision ensures that our conclusions remain focused and directly address the observed genetic interactions. Additionally, we have expanded the Discussion to further clarify the specific nature of *ego-1*(S1198L) with respect to RRF-1.

2. The experiments in Figures 6 and Figure 7C,D are the most important findings of this study, showing that EGO-1 has a role in the licensing of genes important for *exo*-RNAi in the germline (such as *sid-1* and *rde-11*). The apparent persistence of RRF-1-dependent (and presumably HDRE-1-dependent) silencing of *sid-1* and *rde-11* in a genetically wild-type background that correlates with *exo*-RNAi resistance is remarkable, although not novel (it was shown for mutants defective in *P*-granules). The use of *ego-1* missense viable background was instrumental in these experiments. However, it is not clear whether the specific nature of *ego-1*(S1198L) mutation also played a role, such as enhanced production of RRF-1-dependent endogenous silencing small RNAs. The *ego-1*(V1128E) allele is an apparent hypomorph, which is viable and *exo*-RNAi-resistant (Figure 1, EV2A). Performing an experiment shown in Figure 6 with this allele for five generations would be highly illuminating, and either outcome would be interesting.

Thank you for this insightful comment. We agree that investigating whether the specific nature of the *ego-1*(S1198L) mutation contributes to the observed effects is essential. To address this, we performed the experiment shown in Figure 6 using the *ego-1*(V1128E) allele four generations and data is now shown in Fig. EV7.

3. Conclusions from the experiments in Figures 3 and 4 are not convincing. The imaging data can be moved to supplemental materials. The suppression experiments shown in Figure 4A,B are weak. The effects of *cde-1* mutation are hard to interpret, and these data can be omitted. The *znfx-1* and *hrde-1* loss does not affect resistance to *pop-1*. If the authors want to insist on

their model, they should use several additional exo-RNAi target genes producing Emb (or other) phenotypes and repeat the experiments.

Thank you for your valuable feedback. We agree with the concerns raised and have made the suggested changes, including moving the imaging data to Fig. EV4 and omitting the *cde-1* data. Regarding the lack of suppression effects for *pop-1*, we acknowledge the need for further investigation and have performed additional exo-RNAi experiments with target genes *gld-1* (Ste) and *mpk-1* (Ste) to evaluate our model. Both *znfx-1* and *hrde-1* mutants significantly suppressed the Rde phenotype in *ego-1*(S1198L) when subjected to these RNAi, supporting our model. We have added these data in Fig. 3B and EV5A and moved the *pop-1* RNAi data to Fig. EV5B.

*4. The exo-RNAi resistance and reduced *sid-1* and *rde-11* expression correlate. The reduction of these exo-RNAi factors is a plausible explanation for the epigenetic RNAi resistance shown in Figure 6. However, *ego-1*(S1198L); *hrde-1*(-) P0 is resistant to *pop-1*(RNAi) to a large extent (Figure 4B), while *sid-1* and *rde-11* expression is restored in this double compared to single *ego-1*(S1198L) (Figure 5B). Therefore, *ego-1*(S1198L) exo-RNAi resistance is not likely driven to any extent by the misregulation of other RNAi genes. The nature of the (S1198L) mutation is likely to play a major role. Also, surprisingly, *rrf-1*(-) addition to *ego-1*(S1198L) does not restore *sid-1* and *rde-11* expression. Why? The authors do not comment on this.*

Thank you for your detailed comment. To address your concerns, we will incorporate additional experimental data outlined in our response to Comment 3 and revised our description accordingly. Regarding the observation that *rrf-1*(-) addition to *ego-1*(S1198L) does not restore *sid-1* and *rde-11* expression, we hypothesize that this may result from the process by which the *rrf-1* knockout was generated via CRISPR in an *ego-1*(S1198L) mutant background, where *sid-1* and *rde-11* expression was already reduced. This suggests that *rrf-1* may not be required to maintain the reduced expression state once it is established. We will include these points in the revised manuscript.

5. The discussion points about the nature of new EGO-1 missense mutations involving Alpha Fold predictions can be illustrated through Alpha Fold model figures.

Thank you for your comment. We agree that illustrating the discussion points with Alpha Fold model figures would enhance clarity. We included an extended view figure based on Alpha Fold predictions to better visualize the structural implications of the EGO-1 mutations.

*6. The authors should consider a model where *ego-1*(S1198L) affects RRF-1 activity such that it is more active in the endogenous RNAi silencing processes at the expense of exo-RNAi. This could explain the reduced *ts*-sterility in *ego-1*(S1198L), which is RRF-1-dependent, similar to the better-investigated epigenetic inheritance of exo-RNAi resistance. However, the exact*

mechanism of ego-1(S1198L) cannot be explained by genetic methods and is beyond the scope of this study.

Thank you for this insightful and critical comment. We agree that the interaction between ego-1(S1198L) and RRF-1 activity is an important aspect to consider. Based on the results from our additional experiments described above, we discussed about this possibility. We deeply appreciate your suggestion, as it provides valuable direction for interpreting our findings and developing a more comprehensive understanding of the mechanism.

- *Data and the methods are presented in such a way that they can be reproduced.*
- *Statistical analyses are adequate.*

Minor comments:

- *Figure 8C typo: ego-(0) is meant to be shown.*

Thank you for pointing this out. We have updated the allele to ego-1(null) in panel C.

- *Pak and Fire, Science, 2007 should be cited in connection to secondary siRNA production. Ruby and Bartel, Cell, 2006 should be cited as the first study that identified 21U-RNAs.*

Thank you for pointing this out. We added citations to Pak and Fire (Science, 2007) in connection to secondary siRNA production and to Ruby and Bartel (Cell, 2006) as the first study identifying 21U-RNAs.

Reviewer #3 (Significance (Required)):

- *General assessment:*

The strength of this study is in generating reagents suitable for performing experiments that were not feasible with the sterile null mutant. The major finding of the paper is the epigenetic inheritance of resistance to exo-RNAi by the wild-type descendants of ego-1 mutants, which is dependent on rrf-1.

There are numerous weaknesses in the interpretation of other data, which are described in section 1. The study's limitation is the exclusive use of genetic approaches. The effect of the antimorphic point mutations on EGO-1 stability, localization, and interaction with other proteins could have provided more insight into the protein's function.

- *The most notable results presented in the paper are very similar to the findings of several groups published in 2019 (Lev et al., Ouyang et al, and Dodson and Kennedy) and, therefore, are not novel. The experimental setup is identical to Dodson and Kennedy; it just uses different mutants. The novel aspect is the opposite relationship between ego-1 and rrf-1, which has not been described before.*

Full Revision

- *This research will be of interest to C. elegans researchers and those following epigenetic phenomena.*
- *My expertise is in RNAi in C. elegans and epigenetics. I have sufficient expertise to evaluate all aspects of the paper.*

Dear Prof. Mitani,

Thank you for the submission of your revised manuscript to EMBO reports. I contacted referee 3 with it and we have received her/his comments now, which are pasted below. Referee 3 assessed your full point-by-point response to all referee concerns.

As you will see, while the referee acknowledges that the ms has been improved, s/he also notes that a little more experimentation would enhance the study and make it a good fit for EMBO reports. Given that you indicated that you can perform these experiments, I would like to invite you to address these last referee comments and submit a revised ms to EMBO reports.

Please address all referee concerns in a complete point-by-point response. Acceptance of the manuscript will depend on a positive outcome of a second round of review. It is EMBO reports policy to allow a single round of major revision only and acceptance or rejection of the manuscript will therefore depend on the completeness of your responses included in the next, final version of the manuscript.

We realize that it is difficult to revise to a specific deadline. In the interest of protecting the conceptual advance provided by the work, we recommend a revision within 2-3 months (2nd Aug 2025). Please discuss the revision progress ahead of this time with the editor if you require more time to complete the revisions.

- 1) A data availability section providing access to data deposited in public databases is missing. If you have not deposited any data, please add a sentence to the data availability section that explains that.
- 2) Your manuscript contains statistics and error bars based on $n=2$. Please use scatter blots in these cases. No statistics should be calculated if $n=2$.

3) We replaced Supplementary Information with Expanded View (EV) Figures and Tables that are collapsible/expandable online. A maximum of 8 EV Figures can be typeset. EV Figures should be cited as "Figure EV1, Figure EV2" etc... in the text and their respective legends should be included in the main text after the legends of regular figures.

5) a complete author checklist, which you can download from our author guidelines . Please insert information in the checklist that is also reflected in the manuscript. The completed author checklist will also be part of the RPF.

6) Please note that all corresponding authors are required to supply an ORCID ID for their name upon submission of a revised manuscript (). Please find instructions on how to link your ORCID ID to your account in our manuscript tracking system in our Author guidelines

7) Before submitting your revision, primary datasets produced in this study need to be deposited in an appropriate public database (see <https://www.embopress.org/page/journal/14693178/authorguide#datadeposition>). Please remember to provide a reviewer password if the datasets are not yet public. The accession numbers and database should be listed in a formal "Data

Availability" section placed after Materials & Method (see also <https://www.embopress.org/page/journal/14693178/authorguide#datadeposition>). Please note that the Data Availability Section is restricted to new primary data that are part of this study. * Note - All links should resolve to a page where the data can be accessed. *

- the name of the statistical test used to generate error bars and P values,
- the number (n) of independent experiments (please specify technical or biological replicates) underlying each data point,
- the nature of the bars and error bars (s.d., s.e.m.),
- If the data are obtained from n Program fragment delivered error `Can't locate object method "less" via package "than" (perhaps you forgot to load "than"?) at //ejpvfs23/sites23b/embor_www/letters/embor_decision_rc_revise_and_rereview.txt line 56.' 2, use scatter blots showing the individual data points.

12) All Materials and Methods need to be described in the main text using our 'Structured Methods' format, which is required for all research articles. According to this format, the Methods section includes a separate Reagents and Tools Table file (listing key reagents, experimental models, software and relevant equipment and including their sources and relevant identifiers) and a Methods and Protocols section describing the methods using a step-by-step protocol format. The aim is to facilitate adoption of the methodologies across labs. More information on how to adhere to this format as well as a downloadable template (.docx) for the Reagents and Tools Table can be found in our author guidelines: <https://www.embopress.org/page/journal/14693178/authorguide#structuredmethods>.

An example of a Method paper with Structured Methods can be found here: <https://www.embopress.org/doi/full/10.1038/s44320-024-00037-6#sec-4>

I look forward to seeing a newly revised form of your manuscript when it is ready.

Referee #1:

The revised manuscript by Mitani and colleagues has significantly improved the presentation logic and is more focused. The authors were responsive to the reviewers' critiques, and some conclusions, such as suppression of the RNAi resistance phenotype of *ego-1*(S1198L) by the *hrde-1* and *znfx-1* mutations, are now better supported by experimental evidence.

However, the demonstration of the "special" quality of the *ego-1*(S1198L) mutation in inducing "Rde hangover phenotype" as compared to *ego-1*(V1128E), although important, begs for transgenerational experiments with other *ego-1* point mutants described in the paper.

Also, the importance of *sid-1* and *rde-11* downregulation for the special "Rde hangover phenotype" is still uncertain and can now be experimentally tested. Since *ego-1*(S1198L) and the *ego-1* deletion mutant allele *tm521* both lead to reduced *sid-1* and *rde-11* expression (Fig. 4C, D), the special character of *ego-1*(S1198L) could be in something else. A systematic comparison of possible transgenerational RNAi resistance phenotypes in other point mutants and the effects of these mutations on *sid-1* and *rde-11* expression can be performed. Specifically, if *sid-1* and *rde-11* are down in *ego-1*(V1128E), the fact that it does not support the transgenerational phenotype (Fig EV7) would argue against the importance of *sid-1* and *rde-11* regulation for the discovered phenomenon.

I believe that getting additional genetic support for the "special" character of *ego-1*(S1198L), and possibly *ego-1*(R539Q), and the separation from (or stronger connection to) of the discovered transgenerational phenomenon vis-à-vis *sid-1* and *rde-11* regulation, would be valuable and suitable for publication in EMBO Reports. It would be especially novel if it is not related to *sid-1* and *rde-11*, in contrast to the earlier findings with P-granule mutants.

Dear Esther Schnapp,

Thank you very much for your email on May 2 regarding our manuscript (EMBOR-2025-61608V1) titled, "Dual roles of EGO-1 and RRF-1 in regulating germline exo-RNAi efficiency in *Caenorhabditis elegans*". We revised and prepared our manuscript according to your journal's guidelines.

Sincerely yours,

Shohei Mitani, Professor Emeritus, M.D., Ph.D.

Department of Physiology, Tokyo Women's Medical University School of Medicine

8-1, Kawada-cho, Shinjuku-ku, Tokyo, 162-8666, Japan

office +81(352) 69 7362

fax +81(352) 69 7362

mitani.shohei@twmu.ac.jp

Referee #1:

The revised manuscript by Mitani and colleagues has significantly improved the presentation logic and is more focused. The authors were responsive to the reviewers' critiques, and some conclusions, such as suppression of the RNAi resistance phenotype of *ego-1*(S1198L) by the *hrde-1* and *znfx-1* mutations, are now better supported by experimental evidence.

However, the demonstration of the "special" quality of the *ego-1*(S1198L) mutation in inducing "Rde hangover phenotype" as compared to *ego-1*(V1128E), although important, begs for transgenerational experiments with other *ego-1* point mutants described in the paper.

Also, the importance of *sid-1* and *rde-11* downregulation for the special "Rde hangover phenotype" is still uncertain and can now be experimentally tested. Since *ego-1*(S1198L) and the *ego-1* deletion mutant allele *tm521* both lead to reduced *sid-1* and *rde-11* expression (Fig. 4C, D), the special character of *ego-1*(S1198L) could be in something else. A systematic comparison of possible transgenerational RNAi resistance phenotypes in other point mutants and the effects of these mutations on *sid-1* and *rde-11* expression can be performed. Specifically, if *sid-1* and *rde-11* are

down in *ego-1*(V1128E), the fact that it does not support the transgenerational phenotype (Fig EV7) would argue against the importance of *sid-1* and *rde-11* regulation for the discovered phenomenon.

I believe that getting additional genetic support for the “special” character of *ego-1*(S1198L), and possibly *ego-1*(R539Q), and the separation from (or stronger connection to) of the discovered transgenerational phenomenon vis-à-vis *sid-1* and *rde-11* regulation, would be valuable and suitable for publication in EMBO Reports. It would be especially novel if it is not related to *sid-1* and *rde-11*, in contrast to the earlier findings with P-granule mutants.

We thank the reviewer for these insightful suggestions. We have conducted a systematic comparison of potential transgenerational RNAi resistance phenotypes in other point mutants. We have also examined the effect of these *ego-1* MMP mutations on *sid-1* and *rde-11* expression. The comprehensive results of these analyses are now presented in detail in Tables 2 and 3 of the revised manuscript. Our analysis suggests that the penetrance and persistence of the ‘Rde hangover’ phenotype are particularly pronounced in mutants that exhibit strong germline RNAi resistance in the homozygous state.

Dear Prof. Mitani,

Thank you for the submission of your revised manuscript. We have now received the enclosed report from referee 1 who was asked to assess it. Referee 1 only has a few more minor suggestions that I would like you to incorporate before we can proceed with the official acceptance of your manuscript.

A few editorial requests will also need to be addressed:

- The Data Availability Section section should be placed before the Acknowledgments and the second sentence needs to be removed.
- Please correct the conflict of interest subheading to "Disclosure and Competing Interests Statement"
- Please remove the author credits from the ms file.
- Please enter all funding info also in our online ms submission system. NIH Office of Research Infrastructure Programs (P40 OD010440) is currently missing.
- Fig. 3B and the individual panels of Figure 7 (A-D) are missing callouts in the ms text, please add.
- The Reagents & Tools table needs to be removed from the ms file and uploaded separately as a Word file.
- The 3 main tables, Table 1-3, are provided twice; main tables should either be in the ms (between main and EV figures) OR uploaded separately.
- The character and figure count statement should be removed from the ms (title page).

* Figure Legends - Comments *

- Please define the annotated p values ****/***/**/* as well as provide the exact p-values for the same in the legend of figure EV5 A as appropriate.
- Please note that the exact p values are not provided in the legends of figures 1B, C, D, E; 2B, C; 3A, B, D, E, F; 4A-D; 6A, B; EV2 A-D; EV4 C, D; EV6A-E. Please add the exact p-values as reasonable.
- Please indicate the statistical test used for data analysis in the legend of figure EV5 A, B
- Please note that the error bars are not defined in the legends of figures 1B, C, D, E; 2B, C; 3A, B, D, E, F; 4A-D; 5B, C, E, F; 6A, B; C, D; EV1, EV2 A-D; EV4 C, D; EV5A, B; EV6 A-E

I would like to suggest some minor changes to the abstract that needs to be written in present tense. Please let me know whether you agree with this:

RNA interference (RNAi) is widely used in life science research and is critical for diverse biological processes, such as germline development and antiviral defense. In *Caenorhabditis elegans*, RNA-dependent RNA polymerases, with redundant involvement of EGO-1 and RRF-1, facilitate small RNA amplification in germline exogenous RNAi (exo-RNAi). However, their coordination during the regulation of exo-RNAi processes in the germline remains unclear. Here, we examine non-null mutants of the *ego-1* gene and find that *ego-1*(S1198L) animals exhibit germline exo-RNAi defects with normal fertility, abnormalities in germ granules, and synthetic temperature-dependent sterility with *rrf-1*. The exo-RNAi defects in *ego-1*(S1198L) are partially restored by inhibiting *hrde-1* and *znfx-1*. Germline exo-RNAi defects are observed in wild-type and *ego-1*(S1198L) heterozygous descendants derived from *ego-1*(S1198L), but these are suppressed by ancestral inhibition of *rrf-1*. Our data reveal a dual role for EGO-1 in the positive regulation of germline exo-RNAi: it not only mediates target silencing through its RNA-dependent RNA polymerase activity, but also licenses exo-RNAi gene expression, which are antagonized by RRF-1.

EMBO press papers are accompanied online by A) a short (1-2 sentences) summary of the findings and their significance, B) 2-3 bullet points highlighting key results and C) a synopsis image that is exactly 550 pixels wide and 200-600 pixels high (the height is variable). The synopsis image should provide a sketch of the major findings, like a graphical abstract. Please note that text needs to be readable at the final size. Please send us this information along with the final manuscript.

Best regards,

Esther

Referee #1:

The revised version of the manuscript by Mitani and colleagues provides important systematic analyses of the effects of newly discovered ego-1 point mutations on "Rde hangover" phenotype and expression levels of RNAi pathway genes, such as sid-1 and rde-11. These analyses demonstrated that ego-1(R539Q) resembles ego-1(S1198L) both in terms of persistence of "Rde hangover" and reduced expression of rde-11. Although ego-1(R539Q) did not affect sid-1 expression, the authors' suggestion that combined misregulation of many RNAi pathway genes may underly the Rde phenotype is justified. The spatial proximity of R539Q and S1198L further suggests the similarity of molecular perturbations caused by these mutations.

- It is still surprising that rrf-1(-) addition to ego-1(S1198L) does not restore sid-1 and rde-11 expression (Figure 6 A, B), whereas ancestral rrf-1(tm9941) ego-1(S1198L) does (Figure 6 C, D).

The authors' explanation of Figure 6 A, B results on page 15, cited below, does not resolve the problem.

"Given

339 that the rrf-1 knockout was generated via CRISPR in an ego-1(S1198L) mutant
340 background where sid-1 and rde-11 expression was already reduced, this
341 suggests that rrf-1 may not be required to maintain the reduced expression state
342 once it is established".

I suggest removing this sentence and including something like this: Although rrf-1(tm9941) does not restore sid-1 and rde-11 expression when combined with ego-1(S1198L) (Figure 6 A, B), the ancestral rrf-1(tm9941) ego-1(S1198L) facilitates higher sid-1 and rde-11 expression in their rrf-1(+) ego-1(+) G3-G4 descendants compared to ancestral ego-1(S1198L). The molecular mechanisms behind these differences are not clear.

- Results in Figure 3: It is not clear why a minor change in the Nuclear/Cytosolic HRDE-1 ratio is interpreted as significant for HRDE-1 activity. Elevated nuclear HRDE-1 may be relevant, although one can imagine increased HRDE-1 loading with siRNAs without changes in the protein levels. Please amend the discussion of these results.

- Figure EV3 lacks a description of a red channel marker.

- It is not clear if tables 1-3 are meant for the Main text or not. Data for S1198L should be included in Tables 2 and 3 for comparison with R539Q.

- Editing/clarification is required in the examples below, and a thorough reading and editing of the whole manuscript is recommended.

33 but also licenses exo-RNAi gene expression, which "is" antagonized by RRF-1.

49 (Pak & Fire, 2007). In exogenous RNAi (exo-RNAi), the endonuclease dicer "Dicer"

The ego-1 and

77 rrf-1 genes are physically close "in" the genome and are expected to be
78 transcribed together as operons.

89 null mutants exhibit exo-RNAi germline defects. Suggested: "Exhibit defects in exo-RNAi targeting germline genes"

149 showed an Rde phenotype when they were fed with the HT115 expressing
150 germline gene, either pos-1 or pop-1 dsRNA (Table 1 and Fig. 1B, C). "when they were fed HT115 bacteria expressing
dsRNA specific to germline genes"

390 explain how *C. elegans* manages gene expression through different RNAi
Replace with "gene expression regulation"

465 contribute to RNAi defects, along with the RdRP activity of EGO-1, which is
466 directly required for gene silencing.
Replace with "along with reduced RdRP activity of EGO-1"

Referee #1:

The revised version of the manuscript by Mitani and colleagues provides important systematic analyses of the effects of newly discovered *ego-1* point mutations on "Rde hangover" phenotype and expression levels of RNAi pathway genes, such as *sid-1* and *rde-11*. These analyses demonstrated that *ego-1*(R539Q) resembles *ego-1*(S1198L) both in terms of persistence of "Rde hangover" and reduced expression of *rde-11*. Although *ego-1*(R539Q) did not affect *sid-1* expression, the authors' suggestion that combined misregulation of many RNAi pathway genes may underly the Rde phenotype is justified. The spatial proximity of R539Q and S1198L further suggests the similarity of molecular perturbations caused by these mutations.

· It is still surprising that *rrf-1*(-) addition to *ego-1*(S1198L) does not restore *sid-1* and *rde-11* expression (Figure 6 A, B), whereas ancestral *rrf-1*(tm9941) *ego-1*(S1198L) does (Figure 6 C, D).

The authors' explanation of Figure 6 A, B results on page 15, cited below, does not resolve the problem.

"Given

339 that the *rrf-1* knockout was generated via CRISPR in an *ego-1*(S1198L) mutant
340 background where *sid-1* and *rde-11* expression was already reduced, this
341 suggests that *rrf-1* may not be required to maintain the reduced expression state
342 once it is established".

I suggest removing this sentence and including something like this: Although *rrf-1*(tm9941) does not restore *sid-1* and *rde-11* expression when combined with *ego-1*(S1198L) (Figure 6 A, B), the ancestral *rrf-1*(tm9941) *ego-1*(S1198L) facilitates higher *sid-1* and *rde-11* expression in their *rrf-1*(+) *ego-1*(+) G3-G4 descendants compared to ancestral *ego-1*(S1198L). The molecular mechanisms behind these differences are not clear.

We sincerely thank you for this constructive suggestion. We agree that the sentence you proposed more accurately describes the complexity of our findings and candidly acknowledges what remains unclear. It is a significant improvement over our original text. Accordingly, we have replaced the sentence in the manuscript as suggested.

· Results in Figure 3: It is not clear why a minor change in the Nuclear/Cytosolic HRDE-1 ratio is interpreted as significant for HRDE-1 activity. Elevated nuclear

HRDE-1 may be relevant, although one can imagine increased HRDE-1 loading with siRNAs without changes in the protein levels. Please amend the discussion of these results.

Thank you for this important point regarding our interpretation of the data. We agree that our initial interpretation was an overstatement. Following your advice, we have revised this section to avoid over-interpreting the minor change in the Nuclear/Cytosolic ratio and instead focused on the observed fact of overall HRDE-1 accumulation. The revised text now reads: “The overall accumulation of the HRDE-1 protein in the ego-1 (S1198L) mutant suggests that EGO-1 activity is required, directly or indirectly, for maintaining the normal homeostasis of HRDE-1 protein levels. The precise mechanism linking EGO-1’s RdRP function to HRDE-1 protein regulation remains to be determined.” We believe this change makes our discussion more faithful to the data.

- Figure EV3 lacks a description of a red channel marker.

Thank you for pointing out this omission. We have now added a description of the red channel marker to the Figure EV3 legend as follows: “tagRFP::PGL-1 was used as a germ granule marker (red)” .

- It is not clear if tables 1-3 are meant for the Main text or not. Data for S1198L should be included in Tables 2 and 3 for comparison with R539Q.

Thank you for your comment. We sincerely apologize for the confusion regarding Tables 1-3.

You were correct to point this out; the issue was caused by our mistake during submission, where we accidentally uploaded an older, conflicting version of the tables as a separate file in addition to the tables in the main manuscript file. To resolve this, we have now removed the redundant separate file. We believe this has resolved the ambiguity you noted. We also agree that including data for S1198L for comparison with R539Q significantly strengthens our study. As requested, we have now performed the experiments for the S1198L mutant and have included this new data in Tables 2 and 3.

- Editing/clarification is required in the examples below, and a thorough reading and editing of the whole manuscript is recommended.

We are very grateful for your meticulous review and the numerous helpful suggestions for improving the clarity and language of our manuscript. As recommended, we have performed a thorough proofreading and editing of the entire manuscript to improve its readability.

We have incorporated all the specific corrections and suggestions you provided. Below is a point-by-point response:

33 but also licenses *exo*-RNAi gene expression, which "is" antagonized by RRF-1.
Corrected as suggested. Thank you.

49 (Pak & Fire, 2007). In exogenous RNAi (*exo*-RNAi), the endonuclease *dicer* "Dicer"
Corrected to "Dicer" as suggested.

The *ego-1* and

77 *rrf-1* genes are physically close "in" the genome and are expected to be
78 transcribed together as operons.

The preposition "in" has been added as suggested.

89 null mutants exhibit *exo*-RNAi germline defects. Suggested: "Exhibit defects in
exo-RNAi targeting germline genes"

We have revised the sentence as suggested to "Exhibit defects in *exo*-RNAi targeting
germline genes." We appreciate this clearer phrasing.

149 showed an *Rde* phenotype when they were fed with the HT115 expressing
150 germline gene, either *pos-1* or *pop-1* dsRNA (Table 1 and Fig. 1B, C). "when they
were fed HT115 bacteria expressing dsRNA specific to germline genes"

Revised as suggested for improved clarity.

390 explain how *C. elegans* manages gene expression through different RNAi

Replace with "gene expression regulation"

Replaced with "gene expression regulation" as suggested.

465 contribute to RNAi defects, along with the RdRP activity of EGO-1, which is
466 directly required for gene silencing.

Replace with “along with reduced RdRP activity of EGO-1”

Lines 465–466: The phrase has been revised to “along with reduced RdRP activity of EGO-1” as recommended.

Thank you again for your valuable input and for helping us to improve our manuscript.

Dear Dr. Schnapp,

Thank you for the opportunity to revise our manuscript (EMBOR-2025-61608V2). Please find attached the revised version.

We are pleased to inform you that we have now submitted the revised version of our manuscript and all accompanying files through the online portal.

We have carefully addressed all the editorial requests you outlined in your email. Our point-by-point responses to the valuable comments from Referee 1 are also provided in a separate "Response to Reviewer" file, which has been included in the submission.

Regarding the grant P40 OD010440, we would like to clarify that this is not direct funding for our research. This grant supports the Caenorhabditis Genetics Center (CGC), which is funded by the NIH Office of Research Infrastructure Programs. As per their policy, the CGC requests that researchers acknowledge this grant when publishing work that used strains provided by them. For this reason, we have included it in the Acknowledgements section as required, but have not listed it in the author funding section of the submission system. Regarding the Takeda Science Foundation, we've entered "N/A" in the grant number field since no specific grant number was provided.

We believe that the manuscript has been significantly improved, and we hope it is now suitable for publication in your journal.

Thank you again for your time and consideration.

Prof. Shohei Mitani
Tokyo Women's Medical University
Department of Physiology
Kawada-cho
Shinjuku-ku
Tokyo 162-8666
Japan

Dear Prof. Mitani,

I am very pleased to accept your manuscript for publication in the next available issue of EMBO reports. Thank you for your contribution to our journal.

Yours sincerely,
